# Group II truncated haemoglobin YjbI prevents reactive oxygen species-induced protein aggregation in *Bacillus subtilis*

Takeshi Imai[1]\*, Ryuta Tobe[2], Koji Honda[1], Mai Tanaka[2], Jun Kawamoto[3], Hisaaki Mihara[2]\*

[1]Hyogo Prefectural Institute of Technology, Kobe, Japan; [2]Department of Biotechnology, Ritsumeikan University, Shiga, Japan; [3]Institute for Chemical Research, Kyoto University, Kyoto, Japan

**Abstract** Oxidative stress-mediated formation of protein hydroperoxides can induce irreversible fragmentation of the peptide backbone and accumulation of cross-linked protein aggregates, leading to cellular toxicity, dysfunction, and death. However, how bacteria protect themselves from damages caused by protein hydroperoxidation is unknown. Here, we show that YjbI, a group II truncated haemoglobin from *Bacillus subtilis*, prevents oxidative aggregation of cell-surface proteins by its protein hydroperoxide peroxidase-like activity, which removes hydroperoxide groups from oxidised proteins. Disruption of the *yjbI* gene in *B. subtilis* lowered biofilm water repellence, which associated with the cross-linked aggregation of the biofilm matrix protein TasA. YjbI was localised to the cell surface or the biofilm matrix, and the sensitivity of planktonically grown cells to generators of reactive oxygen species was significantly increased upon *yjbI* disruption, suggesting that YjbI pleiotropically protects labile cell-surface proteins from oxidative damage. YjbI removed hydroperoxide residues from the model oxidised protein substrate bovine serum albumin and biofilm component TasA, preventing oxidative aggregation in vitro. Furthermore, the replacement of Tyr[63] near the haem of YjbI with phenylalanine resulted in the loss of its protein peroxidase-like activity, and the mutant gene failed to rescue biofilm water repellency and resistance to oxidative stress induced by hypochlorous acid in the *yjbI*-deficient strain. These findings provide new insights into the role of truncated haemoglobin and the importance of hydroperoxide removal from proteins in the survival of aerobic bacteria.

\*For correspondence:
imai@hyogo-kg.jp (TI);
mihara@fc.ritsumei.ac.jp (HM)

**Competing interest:** The authors declare that no competing interests exist.

## Editor's evaluation

This manuscript is of interest to microbiologists interested in biofilm formation, haem proteins, and cellular responses to extracellular oxidative stress. It proposes a mechanism for biofilm protection from reactive oxygen species (ROIs) through the examination of the Gram positive, *Bacillus subtilis*. The results highlight the importance of protein peroxidase activity for proper biofilm assembly in a model species.

## Introduction

Truncated haemoglobins (trHbs) are small haem proteins found in microbes and plants (***Wittenberg et al., 2002***; ***Boechi et al., 2010***). They belong to the globin superfamily and form a distinct family separated from bacterial flavohemoglobins, *Vitreoscilla* haemoglobin, plant symbiotic and non-symbiotic haemoglobins, and animal haemoglobins (***Vuletich and Lecomte, 2006***; ***Boechi et al., 2010***). The haem group enables trHbs to bind to small biomolecules, such as oxygen (O₂), carbon monoxide (CO),

and nitric oxide (NO) (**Wittenberg et al., 2002**). Additionally, haem-pocket residues are highly structurally variable, conferring trHbs with diverse roles, such as scavengers (**Ouellet et al., 2002**) or transport carriers (**Liu et al., 2004**; **Wittenberg et al., 2002**) of the corresponding small ligands. Such versatility is possibly related to defence mechanisms against the harmful effects of reactive oxygen species (ROS). trHbs are also implicated in the pathogenicity of some bacteria (**Pawaria et al., 2007**; **Ascenzi et al., 2008**; **Wittenberg et al., 2002**), but the molecular functions of these proteins are unclear.

The trHb family proteins are further divided into three subfamilies, Groups I (trHbN), II (trHbO), and III (trHbP), with each subfamily having different structural characteristics (**Busta-mante et al., 2016**; **Wittenberg et al., 2002**). Among these subfamilies, trHbO proteins evolu-tionarily originated before trHbN and trHbP proteins and are distributed into Actinobacteria, Proteobacteria, Firmicutes, and plants (**Vuletich and Lecomte, 2006**; **Boechi et al., 2010**). A trHbO from *Mycobacterium tuberculosis* exhibits inefficient gas-exchange capacity with a very slow release of $O_2$ (**Ouellet et al., 2003**) and a slower reaction rate with NO than other trHb subfamilies (**Ouellet et al., 2003**), suggesting that trHbOs play additional roles other than trans-porting these molecules (**Giangiacomo et al., 2005**). In accordance, trHbOs from *M. tubercu-losis* (**Ouellet et al., 2007**), *Thermobifida fusca*

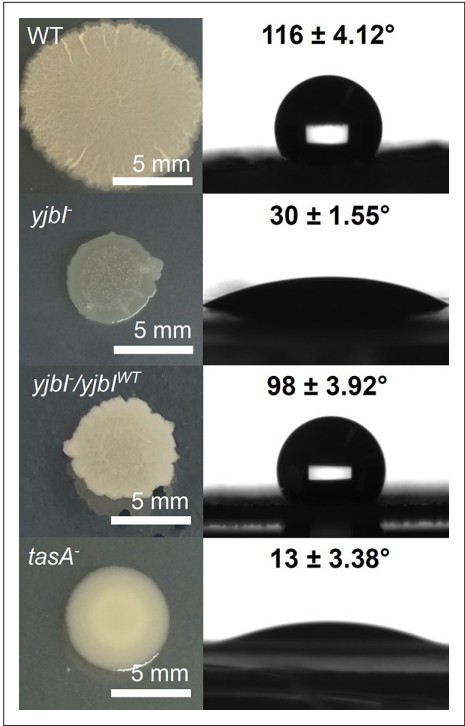

**Figure 1.** YjbI is needed for normal biofilm formation. Images showing the morphology (left panels) and surface repellency (right panels) of the biofilms of the wild-type (WT), *yjbI⁻*, *yjbI⁻/yjbIᵂᵀ*, and *tasA*-deficient (*tasA⁻*) strains of *Bacillus subtilis*. The water contact angles indicated in the right panels represent the mean ± SD of three independent experiments (n=3).

(**Torge et al., 2009**), and *Roseobacter denitrificans* (**Wang et al., 2015**) can exhibit peroxidase activity on hydrogen peroxide in vitro. However, because these bacteria express catalases that scavenge hydrogen peroxide (**Manca et al., 1999**), the physiological relevance of the peroxidase activity of trHbOs remains unknown.

The Gram-positive bacterium *Bacillus subtilis* possesses a trHbO, termed YjbI. YjbI has been heter-ologously expressed as a recombinant protein in *Escherichia coli*, and the purified protein has been characterised (**Choudhary et al., 2005**). It exists as a monomer with a molecular mass of 15.1 kDa and comprises 132 amino acid residues (**Choudhary et al., 2005**). The three-dimensional crystal structure of YjbI in the form of the cyano-Met derivative was determined at 2.15 Å resolution (**Giangiacomo et al., 2005**), revealing intriguing structural features, including a shallow depression on the proximal side of the haem. The binding properties of YjbI to $O_2$ (**Boechi et al., 2010**; **Ouellet et al., 2002**), CO (**Boechi et al., 2010**; **Ouellet et al., 2002**; **Choudhary et al., 2005**; **Feis et al., 2008**; **Lapini et al., 2012**), and hydrogen sulphide **Nicoletti et al., 2010** have been extensively studied in vitro. Some have observed that YjbI has a high affinity for $O_2$ and hydrogen sulphide, and the oxygenated deriv-ative is significantly stable, which may rule out a role of YjbI in $O_2$ transport/storage. As is common for many haem proteins, an inherent low peroxidase activity of YjbI on hydrogen peroxide has been reported (**Choudhary et al., 2005**), but this observation remains controversial. The *yjbI* gene is part of the *yjbIH* operon. YjbH is a bacterial adaptor protein required for efficient degradation of the disulphide-stress-activated transcription regulator Spx by the protease ClpXP (**Larsson et al., 2007**; **Awad et al., 2019**). Although these previous studies have implied a possible link between YjbI and response to oxidative stress, the exact function of YjbI remains unclear.

In this study, we present in vivo and in vitro analyses of the physiological function of *B. subtilis* YjbI. We show *yjbI* is required for the formation of biofilms with water repellence. YjbI was found to

suppress the ROS-mediated aggregation of the major biofilm matrix protein TasA and localise to the bacterial cell surface or the biofilm matrix. We show that, besides a role in biofilm maintenance, YjbI functions as an antioxidant protein. Moreover, we observed YjbI exerted a peroxidase-like activity on a protein hydroperoxide substrate. YjbI is proposed to function as an antioxidant protein in protecting cell-surface proteins from ROS-induced aggregation via its protein hydroperoxide peroxidase-like activity.

## Results

### *yjbI* is required for normal formation of the *B. subtilis* biofilm

We observed unusual biofilm formation by the *yjbI*-deficient mutant on a solid medium. Specifically, the *B. subtilis* 168 wild-type (WT) strain formed architecturally complex colonies with characteristic wrinkles on the MSgg solid medium, whereas the *yjbI*-deficient mutant failed to establish such biofilms; instead, it formed flat and glossy colonies (*Figure 1a*). Because of the water-repellent properties of the mature biofilm surfaces, the functional integrity of a biofilm can be inferred from the contact angle of a water droplet placed on top of the biofilm (*Arnaouteli et al., 2017*; *Arnaouteli et al., 2016*; *Kobayashi and Iwano, 2012*). Through this method, we examined the integrity of the biofilms of the *yjbI*-deficient mutant. Water droplets placed on the surfaces of the WT biofilms assumed nearly spherical shapes with a contact angle of 116±4.12°, indicating the remarkable surface repellence of the WT biofilms (*Figure 1a*). In contrast to the WT strain, the water droplets on the surfaces of *yjbI*-deficient mutant biofilms had a contact angle of 30±1.55°, indicating a significant loss of surface repellence in the deficient mutant biofilms. Similarly, the *tasA*-deficient mutant (COTNd), which was a negative control lacking TasA, the main fibrous protein in *B. subtilis* biofilms (*Diehl et al., 2018*; *Hobley et al., 2015*; *Romero et al., 2010*), failed to form biofilms with normal wrinkles and water repellence (*Figure 1a*). When *yjbI* was reintroduced into the *yjbI*-deficient mutant, the ability of the strain to form biofilms with wrinkles and surface repellence was markedly recovered (*Figure 1a*). These results indicated that YjbI is required for normal biofilm formation.

### YjbI suppresses ROS-mediated TasA aggregation

Since the *yjbI*-deficient mutant lost the biofilm surface repellence in a manner like that of the *tasA*-deficient mutant, we examined the effect of *yjbI* deficiency on TasA. His-tagged TasA (TasA-His) was expressed in the WT and *yjbI*-deficient strains carrying pHtasA1 by cultivating in the biofilm-promoting liquid MSgg medium, and the soluble and insoluble protein fractions from the lysate of the pellicles of each strain were analysed by SDS-PAGE followed by western blotting using an anti-His-tag antibody. A band with the expected molecular mass of 31 kDa for TasA-His was detected in the soluble fractions from both the WT and *yjbI*-deficient strains (*Figure 2a*). The 31 kDa TasA-His band was also detected in the insoluble fraction from the WT strain (*Figure 2a*). In contrast, we observed smeared bands with markedly high molecular masses (>225 kDa) in the insoluble fraction from the *yjbI*-deficient mutant (*Figure 2a*), suggesting that YjbI prevents TasA from forming high-molecular-mass aggregates. No monomeric TasA was detected in the insoluble fraction of the *yjbI*-deficient mutant strain. An aggregate of TasA was observed under strong reducing and heat-denaturing conditions in SDS sample buffer, suggesting that covalent bonds may be involved in aggregate formation.

Covalent cross-linking of proteins is generally known to be caused by ROS-mediated protein oxidation (*Davies, 2016*; *Gebicki, 1997*; *Dean et al., 1997*). We could partly reproduce the TasA aggregation in vitro by oxidising purified TasA$_{28-261}$-His (a mature form of TasA with a C-terminal His-tag) at neutral pH in the presence of hydrogen peroxide and Fe$^{2+}$, which undergo the Fenton reaction (*Gebicki, 1997*; *Welch et al., 2002*). The Fenton reaction produces hydroxyl radicals according to the following equation: Fe$^{2+}$ + H$_2$O$_2$→ Fe$^{3+}$ + OH$^-$ + ·OH, and was used to induce ROS-mediated protein oxidation. We investigated whether YjbI could prevent this oxidation-induced aggregation and found that adding purified YjbI (*Figure 2—figure supplement 1*) to the mixture significantly suppressed the formation of TasA$_{28-261}$-His aggregates (*Figure 2b*), consistent with the results of the in vivo experiment shown in *Figure 2a*. These findings suggest that YjbI suppresses ROS-mediated TasA aggregation.

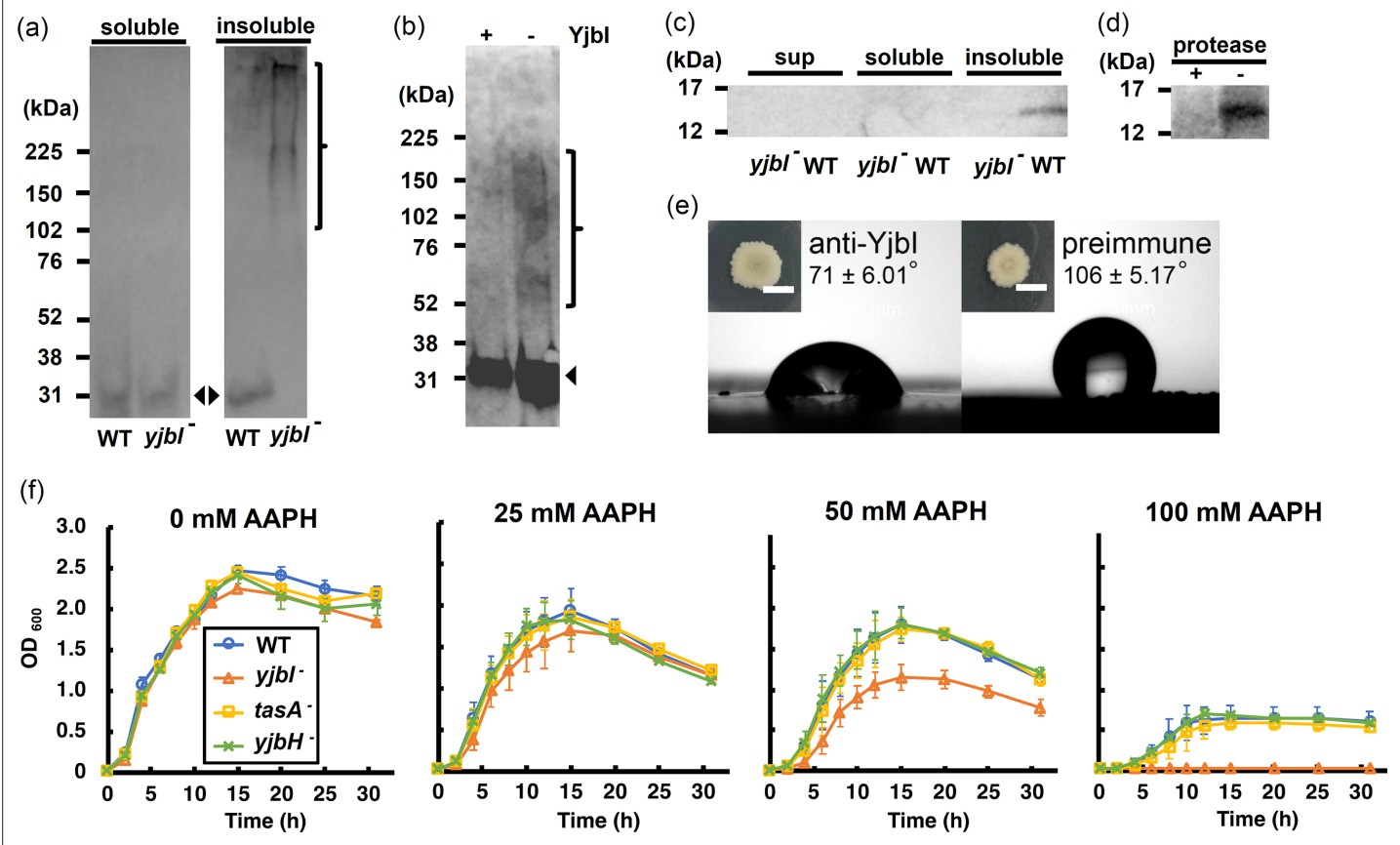

**Figure 2.** *yjbI* is required to prevent oxidative TasA aggregation and for the normal antioxidant properties of *Bacillus subtilis* cells. (**a**) Detection of His-tagged TasA (TasA-His) in the soluble (left panel) and insoluble (right panel) fractions of the pellicles of the wild-type (WT) strain or *yjbI*-deficient mutant (*yjbI*) strain carrying pHtasA1, which constitutively expresses TasA-His. After culturing in liquid MSgg medium, lysates of these pellicles were further analysed by western blotting using an anti-His-tag antibody. The positions of the molecular mass marker proteins are shown on the left. The arrowheads and bracket indicate monomeric and aggregated TasA-His, respectively. Similar results were obtained from three independent experiments. (**b**) In vitro reactive oxygen species (ROS)-induced formation of aggregates of purified TasA (pEtasA2-derived TasA$_{28-261}$-His) in the presence (+) or absence (-) of purified recombinant YjbI, as evidenced by western blotting analysis using an anti-His-tag antibody. The arrowhead and bracket indicate monomeric and aggregated TasA$_{28-261}$-His, respectively. Protein aggregation was induced with 1 mM H$_2$O$_2$ and 10 µM FeCl$_2$. Similar results were obtained from two independent experiments. (**c**) Localisation of YjbI in the insoluble biofilm fraction. The soluble and insoluble fractions of the WT and *yjbI* pellicles and culture supernatants (sup) were analysed by western blotting using an anti-YjbI antiserum. Similar results were obtained from two independent experiments. (**d**) Localisation of YjbI to the cell surface, as evidenced by western blot analysis of the intact WT pellicles with (+) or without (-) protease digestion. YjbI was detected using an anti-YjbI antiserum. Similar results were obtained from two independent experiments. (**e**) Images showing biofilm surface repellency. Two microliters of rabbit anti-YjbI serum (left panel) or pre-immune serum (right panel) were mixed with 2 µL of pre-cultured *Bacillus subtilis* cells and inoculated on MSgg solid medium. The morphologies of the biofilms are shown in the insets. The water contact angles indicated in the right panels represent the mean ± SD of three independent experiments. (**f**) Sensitivity of planktonically grown *B. subtilis* strains to 2,2'-azobis(2-amidinopropane) dihydrochloride (AAPH)-induced oxidative stress. *B. subtilis* cells (WT, *yjbI*-, *tasA*-, or *yjbH*-) were inoculated at OD$_{600}$=0.02 in LB medium containing 0–100 mM AAPH and grown at 37°C with shaking. Data are shown as the mean ± SD of three independent experiments.

The online version of this article includes the following source data and figure supplement(s) for figure 2:

**Source data 1.** PVDF membrane after western blotting treatment of soluble and insoluble fractions of *Bacillus subtilis* extracts with anti-TasA antibody.

**Source data 2.** Gel image of Fenton reaction-treated TasA after electrophoresis and silver staining.

**Source data 3.** PVDF membrane after western blotting treatment of each fraction of *Bacillus subtilis* using anti-YjbI antiserum.

**Source data 4.** PVDF membrane after western blotting treatment of protease-treated *Bacillus subtilis* extracts using anti-YjbI antiserum.

**Figure supplement 1.** Purification of YjbI.

**Figure supplement 1—source data 1.** Gel image of column-purified recombinant YjbI after electrophoresis.

## YjbI localises to the cell surface or the biofilm matrix of *B. subtilis*

TasA localises to the cell surface by the Sec pathway (*Romero et al., 2010*; *Romero et al., 2011*). Because YjbI suppressed TasA aggregation in vivo (*Figure 2a*), it is reasonable to expect that YjbI also localises to the cell surface. To address this hypothesis, the soluble and insoluble fractions of cell lysates from WT and *yjbI*-deficient *B. subtilis* pellicles and the planktonically grown culture supernatants were analysed by western blotting using an anti-YjbI antiserum. YjbI was detected only in the insoluble fraction from the WT pellicles (*Figure 2c*). We then examined whether YjbI is exposed on the cell surface. Towards this end, we treated intact *B. subtilis* WT pellicles with a protease cocktail before western blot analysis of the whole cells. The protease treatment almost eliminated the immunoreactive YjbI from the intact *B. subtilis* WT pellicles, suggesting that YjbI is localised to the extracellular surface (*Figure 2d*). Furthermore, we examined whether YjbI could be recognised by anti-YjbI antibodies, which must be cell impermeable. Treatment of the intact WT cells with the anti-YjbI antiserum before cultivation on MSgg medium impaired colony biofilm formation, with a significant loss of surface repellence (*Figure 2e*), as observed for the *yjbI*-deficient mutant. In contrast, treatment with the control pre-immune serum showed almost no effect. These observations suggest that anti-YjbI antibodies can interact with YjbI to impair its function on the biofilm surface. Taken together, these results suggest that YjbI localises to the cell surface or the biofilm matrix of *B. subtilis*.

## YjbI functions as an antioxidant protein in *B. subtilis*

To address whether YjbI functions exclusively in biofilm maintenance or in a general cellular protection against oxidative stress, we analysed the effect of 2,2'-azobis(2-amidinopropane) dihydrochloride (AAPH)-induced oxidative stress on the planktonic growth of *B. subtilis* strains in lysogeny broth (LB) liquid medium under shaking conditions. AAPH was chosen as the radical initiator because it gently generates free radicals under neutral pH (*Gebicki and Gebicki, 1993*) and is suitable for a long-term cultivation experiment. We observed no significant difference between the WT strain and *tasA*-deficient mutant in the sensitivity to AAPH, indicating that TasA plays no apparent role in the protection of the planktonically grown cells from oxidative stress (*Figure 2f*). In contrast, the *yjbI*-deficient mutant showed hypersensitivity to AAPH compared with the WT and *tasA*-deficient mutant (*Figure 2f*). This result shows that the hypersensitivity of the *yjbI*-deficient mutant to AAPH is not because of TasA impairment in this mutant but presumably because YjbI is involved in a general cellular protection against AAPH-induced oxidative stress, at least during the planktonic growth. Interestingly, disruption of *yjbH* (BKE11550, ), which is co-transcribed with *yjbI* (*Rogstam et al., 2007*), did not alter the sensitivity to AAPH-induced oxidative stress (*Figure 2f*), suggesting that YjbI functions independently of YjbH under oxidative stress.

The effect of the oxidative stress induced by hypochlorous acid (HClO) on the *yjbI*-deficient mutant was also examined. HClO is a strong bactericidal agent that can cause severe protein hydroperoxidation via the generation of radicals on amino acid residues (*Hawkins and Davies, 1998*; *Stadtman and Levine, 2003*). A minimum bactericidal concentration assay revealed that the sensitivity of the *yjbI*-deficient mutant to hypochlorous acid was approximately 100 times higher than the WT strain (*Table 1*). These data demonstrate that the loss of *yjbI* increases the sensitivity of the cells to oxidative stress, suggesting that YjbI functions as a powerful antioxidant protein in *B. subtilis*. Taken together, these results suggest that the antioxidant role of YjbI is not limited to maintaining the function of TasA in biofilm formation; most likely, YjbI functions to protect any labile cell-surface protein from oxidative damage.

**Table 1.** Minimum bactericidal concentration following exposure to hypochlorous acid (HClO) (n=2).

| HClO (mM) | 62.5 | 31.3 | 15.6 | 5.00 | 2.50 | 1.25 | 0.625 | 0.313 | 0.156 |
|---|---|---|---|---|---|---|---|---|---|
| WT | − | + | + | + | + | + | + | + | + |
| *yjbI⁻* | − | − | − | − | − | − | − | + | + |

(+) visible growth, (−) no visible growth.

## YjbI exhibits protein hydroperoxide peroxidase-like activity

The observed ROS-mediated TasA aggregation likely involves intermolecular covalent cross-linking among individual TasA proteins (*Figure 2a and b*). Protein cross-links are formed via radical reactions and the Michael additions after the generation of protein hydroperoxides (half-life: ~4 hr at 37°C) (*Davies, 2016*), (*Gieseg et al., 2000*; *Figure 3a*). Therefore, we examined in vitro whether YjbI could prevent the ROS-mediated aggregation of bovine serum albumin (BSA), which has been a model protein in studies of protein hydroperoxides (*Gebicki and Gebicki, 1993*). A hydroperoxidised BSA (BSA-OOH) was prepared using the Fenton reaction followed by removal of excess hydrogen peroxide and unstable radicals via cold acetone precipitation (*Gieseg et al., 2000*). Subsequent incubation of BSA-OOH in the absence of YjbI promoted the spontaneous self-crosslinking of BSA to yield BSA aggregates and a concomitant fragmentation via nonspecific free radical chain reactions (*Figure 3a*; *Figure 3b*). In contrast, the addition of YjbI to the subsequent incubation of BSA-OOH markedly suppressed the BSA-OOH aggregation (*Figure 3b*). An identical result was obtained for BSA-OOH prepared using AAPH (*Figure 3c*). The protective function against oxidative BSA aggregation/fragmentation was specific to YjbI, because two typical haem proteins, haemoglobin and myoglobin, showed no protective effects, but promoted the oxidative BSA aggregation/fragmentation (*Figure 3d*). The effects of haemoglobin and myoglobin are consistent with a previous observation that coexistence of a haem group and an oxidant generally accelerates radical reactions (*Svistunenko, 2005*). Time course analysis further demonstrated the rapid and significant elimination of the hydroperoxide groups from BSA-OOH by YjbI (*Figure 3e*). To provide direct experimental evidence of the proposed link between TasA and YjbI, we purified mature TasA (*Figure 3—figure supplement 1*) and prepared TasA-OOH using the same procedure as for BSA-OOH. The addition of YjbI to TasA-OOH significantly decreased its hydroperoxide groups (*Figure 3f*), as observed in BSA-OOH. Taken together, these results suggest that YjbI prevents protein aggregation/fragmentation, most likely through its haem-mediated protein hydroperoxide peroxidase-like activity (*Figure 3a*).

## Effect of amino acid substitution in YjbI on its in vivo and in vitro functions

We investigated whether phenotypic abnormalities in biofilms were related to the structure near the possible active centre of YjbI. Point mutations were introduced into electron-rich residues near the haem of YjbI to produce its derivatives, Y25F, Y63F, and W89F. Each plasmid expressing one of these derivatives was introduced into a *yjbI*-deficient strain and examined for biofilm integrity. The results showed that water repellence in the *yjbI*-deficient strain was not rescued by the introduction of Y63F, while Y25F and W89F complemented the *yjbI*-deficient phenotype (*Figure 4a*). This result indicates that Tyr[63] is important for YjbI function. Furthermore, HClO resistance was partially restored by the introduction of a plasmid-encoded WT *yjbI* gene (*yjbI*[WT]) into the *yjbI*-deficient strain (*Table 2*). Among the mutant strains complemented with the *yjbI* derivatives, *yjbI*[-]/*yjbI*[W89F] strain recovered its HClO resistance to the same extent as the *yjbI*[-]/*yjbI*[WT], whereas both *yjbI*[-]/*yjbI*[Y25F] and *yjbI*[-]/*yjbI*[Y63F] mutants showed high sensitivity to HClO, similar to the *yjbI*-deficient mutant. The apparent discrepancy between the biofilm water repellence and the HClO sensitivity of *yjbI*[-]/*yjbI*[Y25F] may be due to differences in experimental conditions, such as the levels of oxidative stress and the type of growth (biofilms or planktonic). To determine whether the significant phenotypic changes observed for *yjbI*[-]/*yjbI*[Y63F] relate to the protein hydroperoxide peroxidase-like activity of YjbI, we purified the Y63F derivative (*Figure 4—figure supplement 1*) and evaluated its ability to remove the hydroperoxide groups in BSA-OOH. We found that Y63F lost its hydroperoxide peroxidase-like activity (*Figure 4b*). Tyr[63] was also conserved in *M. tuberculosis* trHbO, and this residue has been predicted to interact with haem most significantly (*Ouellet et al., 2007*). Our results clearly indicate that Tyr[63] is important for both the in vivo and in vitro functions of YjbI, suggesting that its protein hydroperoxide peroxidase-like activity is linked to the maintenance of biofilm integrity.

## YjbI does not affect lipid peroxidation

Lipids, one of the cell-surface components, are also hydroperoxidised by ROS (*Cabiscol et al., 2000*). We examined whether *yjbI* disruption also affects the amount of lipid hydroperoxides in biofilms. The colony biofilms of the WT and *yjbI*-deficient strains were prepared on solid MSgg medium, and the amounts of lipid hydroperoxide in the collected biofilms were determined. Therefore, the level of

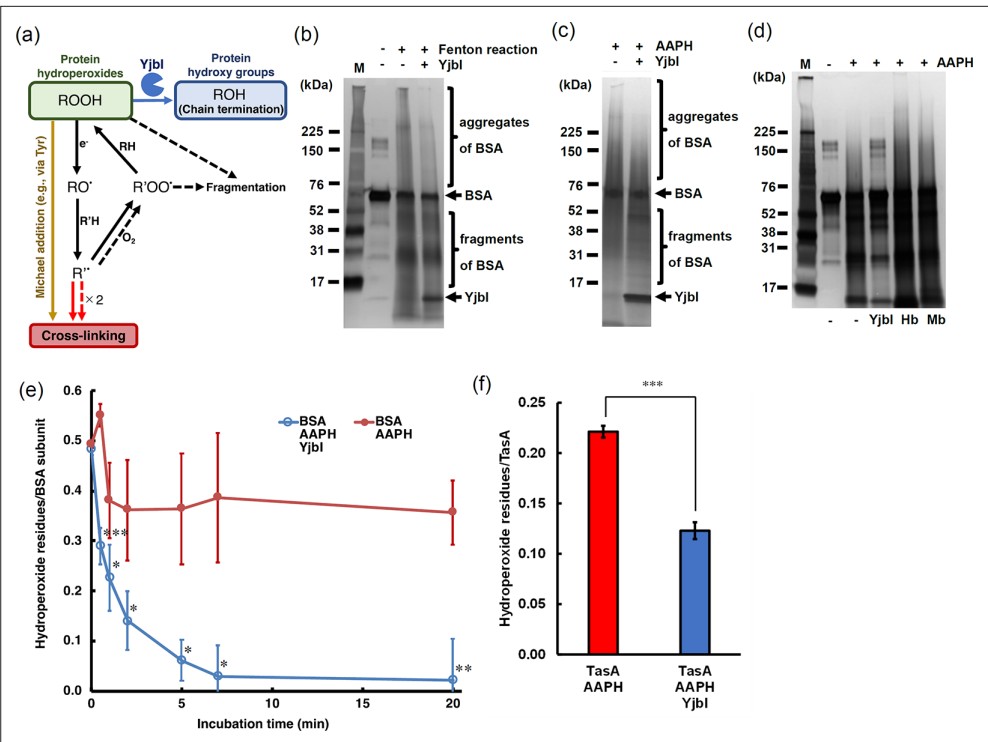

**Figure 3.** YjbI exhibits protein hydroperoxide peroxidase activity. (**a**) Schematic drawing summarising the proposed protein cross-linking and fragmentation induced by protein radicals derived from protein hydroperoxides and prevention of these reactions by the protein hydroperoxide peroxidase activity of YjbI. The solid and dashed arrows indicate the already known reactions (***Davies, 2016***) occurring at protein side chains and backbones, respectively. Protein cross-linking proceeds via spontaneous radical coupling (red arrows), and the Michael addition occurs at protein side chains (yellow arrow). YjbI catalyses the protein hydroperoxide peroxidase reaction (blue arrow) to convert protein hydroperoxides (ROOH) to hydroxy groups (ROH) at protein side chains and prevents further protein cross-linking and fragmentation. Side chain alkoxyl radicals (RO·) can be formed through one-electron reduction (e.g., homolysis) of ROOH, followed by hydrogen atom abstraction reactions with other protein side chains or backbones (R'H). The carbon-centred radicals (R'·) of side chains or backbones lead to protein cross-linking to yield protein aggregates (indicated with solid and dashed red arrows). The side chain or backbone R'· also reacts with oxygen to produce protein peroxy radicals (R'OO·) under aerobic conditions. The side chain R'OO· reacts with the side chains or backbones of other amino acid residues (RH) in its vicinity to yield protein ROOH moieties. The unstable backbone R'OO· and ROOH moieties cause protein fragmentation. (**b**) BSA was treated with (+) or without (-) $H_2O_2$ and $FeCl_2$ to produce BSA-OOH by Fenton reaction. BSA-OOH was incubated in the presence (+) or absence (-) of YjbI and analysed by SDS-PAGE and silver staining. YjbI and monomeric BSA are indicated by arrows, and BSA aggregates and fragments are indicated by brackets. The sizes of the molecular mass marker proteins (**M**) are shown on the left. Similar results were obtained from two independent experiments. (**c**) BSA-OOH prepared by 2,2′-azobis(2-amidinopropane) dihydrochloride (AAPH) treatment was incubated in the presence (+) or absence (-) of YjbI and analysed by SDS-PAGE and silver staining. The labels on the left or right of the images are the same as those in (**b**). Similar results were obtained from two independent experiments. (**d**) YjbI prevents AAPH-induced BSA-OOH aggregation. BSA was incubated for 3 hr with (+) or without (-) AAPH before adding YjbI, haemoglobin from bovine blood (Hb) (Sigma-Aldrich), myoglobin from equine heart (Mb) (Sigma-Aldrich), or none (-) and then analysed by SDS-PAGE and silver staining. The positions of the molecular mass marker proteins (**M**) are shown on the left. Similar results were obtained from two independent experiments. (**e**) Time course of the *YjbI*-induced reduction of the hydroperoxide groups in BSA-OOH. Peroxide-treated BSA (30 µM) in 50 mM Tris-acetate buffer (pH 7.0) was incubated with (blue circles) or without (red circles) YjbI (3.3 µM) at 37°C, and the reaction was terminated by adding 4 volumes of cold acetone at the indicated incubation times (0, 0.5, 1, 2, 5, 7, and 20 min). The number of hydroperoxide groups per BSA subunit was determined for each sample. The data are the mean ± SD of three independent experiments (*p<0.05; **p<0.01; and ***p<0.005; *t*-test, one-tailed). (**f**) TasA-OOH (30 µM) in 50 mM Tris-acetate buffer (pH 7.0) was incubated with YjbI (3.3 µM) at 37°C, and the reaction was terminated by adding 4 volumes of cold acetone after

*Figure 3 continued on next page*

*Figure 3 continued*

incubation for 20 min. The number of hydroperoxide groups per TasA was determined for each sample. Data are shown as the mean ± SD of three independent experiments (***p<0.005; *t*-test, one-tailed).

The online version of this article includes the following source data and figure supplement(s) for figure 3:

**Source data 1.** Gel image of Fenton reaction-treated BSA after electrophoresis and silver staining.

**Source data 2.** Gel image of 2,2'-azobis(2-amidinopropane) dihydrochloride (AAPH)-treated BSA after electrophoresis and silver staining.

**Source data 3.** Gel image of 2,2'-azobis(2-amidinopropane) dihydrochloride (AAPH)-treated BSA and haem proteins after electrophoresis and silver staining.

**Figure supplement 1.** Purification of TasA and YjbI (Y63F).

**Figure supplement 1—source data 1.** Gel image of column-purified recombinant TasA after electrophoresis.

**Figure supplement 2.** Lipid hydroperoxide (LOOH) levels were not affected in *yjbI-* strain biofilms.

**Figure supplement 3.** X-ray crystal structure of *Bacillus subtilis* YjbI (PDB ID: 1UX8) (*Giangiacomo et al., 2005*), showing an opening on the molecular surface.

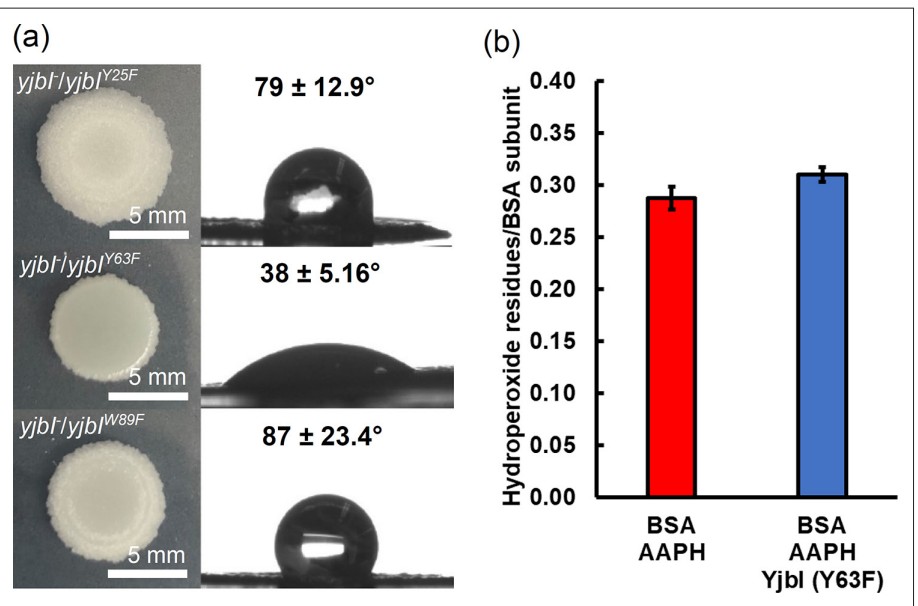

**Figure 4.** Gene complementation study using the *yjbI*-deficient strain of *Bacillus subtilis* with plasmids expressing YjbI derivatives (Y25F, Y63F, and W89F). **a)** The morphology and surface repellence of the biofilms of the *yjbI*-deficient strains carrying each plasmid are shown. The water contact angles indicated in the right panels represent the mean ± SD of three independent experiments (n=3). (**b**) BSA-OOH (30 µM) in 50 mM Tris-acetate buffer (pH 7.0) was incubated with the YjbI derivative Y63F (3.3 µM) at 37°C, and the reaction was terminated by adding 4 volumes of cold acetone after incubation for 20 min. The number of hydroperoxide groups per BSA subunit was determined for each sample. Data are shown as the mean ± SD of three independent experiments.

The online version of this article includes the following source data and figure supplement(s) for figure 4:

**Figure supplement 1.** Y63F derivative of YjbI was produced in *Escherichia coli* BL21(DE3)/pEyjbI3 cells grown in lysogeny broth (LB) medium.

**Figure supplement 1—source data 1.** Gel image of column-purified recombinant YjbI (Y63F) after electrophoresis.

**Table 2.** Minimum bactericidal concentration following exposure to hypochlorous acid in the *yjbI⁻/yjbI*$^{WT}$ and point-mutated strains (HClO) (n=2).

| HClO (mM) | 62.5 | 31.3 | 15.6 | 5.00 | 2.50 | 1.25 | 0.625 | 0.313 | 0.156 |
|---|---|---|---|---|---|---|---|---|---|
| *yjbI⁻/yjbI*$^{WT}$ | − | − | − | − | − | + | + | + | + |
| *yjbI⁻/yjbI*$^{Y25F}$ | − | − | − | − | − | − | − | + | + |
| *yjbI⁻/yjbI*$^{Y63F}$ | − | − | − | − | − | − | − | + | + |
| *yjbI⁻/yjbI*$^{W89F}$ | − | − | − | − | − | ± | + | + | + |

(+) visible growth, (−) no visible growth, (±) visible growth in one of two cases.

lipid hydroperoxide was not significantly increased in the *yjbI*-deficient mutant relative to the level in the WT, implying that YjbI does not function in eliminating lipid hydroperoxides (*Figure 3—figure supplement 2*).

## Discussion

In this study, we revealed a physiological and biochemical function of YjbI. Our results suggest that YjbI prevents the ROS-induced aggregation of proteins, such as TasA, localised on the cell surface of *B. subtilis*. We also showed that YjbI may possess a unique protein hydroperoxide peroxidase-like activity, absent from other haem proteins, such as haemoglobin and myoglobin. Oxidative protein aggregation and protein carbonyls derived from protein radicals and hydroperoxides *Davies, 2016* have emerged as important biomarkers of various cellular defects caused by oxidative stress not only in mammals (*Korovila et al., 2017*; *Heinecke et al., 1993*) but also in bacteria (*Ling et al., 2012*). Peroxiredoxins have been reported to repair intracellular protein peroxidation in mammals (*Peskin et al., 2010*). However, YjbI is distinct from peroxiredoxins in that it is a haem protein with no significant sequence homology (<15%). The second-order rate constants ($M^{-1} \cdot s^{-1}$) for the reactions of mammalian peroxiredoxins 2 and 3 with BSA-OOH are 160 and 360, respectively, and have been shown to reduce protein hydroperoxides more efficiently than GSH under physiological conditions (*Peskin et al., 2010*). Although direct comparison is difficult due to different experimental conditions, YjbI and peroxiredoxins are likely to have a similar catalytic rate, as both proteins can reduce BSA-OOH in the order of several µM in roughly 5 min at similar protein concentrations (*Figure 3e*; *Peskin et al., 2010*). Interestingly, selenomethionine can catalyse the removal of hydroperoxides from proteins in the presence of GSH or a thioredoxin system (*Suryo Rahmanto and Davies, 2011*). However, this system, as well as peroxiredoxins, localises in the cytoplasm of cells, which is a significant difference between YjbI and these proteins. Moreover, whether bacteria utilise peroxiredoxins and the selenomethionine system to remove hydroperoxides from proteins remains unclear.

Most studies on biofilm formation in *B. subtilis* use the *B. subtilis* NCBI3610 strain as a model bacterium because of its ability to form well-structured three-dimensional biofilms (*Arnaouteli et al., 2021*; *Mielich-Süss and Lopez, 2014*). The biofilms of the WT and *tasA* mutant strains of the *B. subtilis* 168 strain are known to be morphologically different from those of the *B. subtilis* NCBI3610 strain (*Romero et al., 2010*; *Vlamakis et al., 2008*; *Erskine et al., 2018*). In this study, the *B. subtilis* 168 strain was used because it is the most representative of *B. subtilis* and serves as a model organism for a wider range of research aspects (*Zeigler et al., 2008*) as we were not only interested in evaluating biofilm formation but also in more general aspects of oxidative damage responses in bacteria.

In *B. subtilis*, several cytosolic factors have been known to protect the cells from oxidative damages. The catalase KatA (*Chen et al., 1995*) and hydrogen peroxide peroxidase AhpC (*Broden et al., 2016*) protect the cells from hydrogen-peroxide-induced oxidative damages. OhrA and OhrB contribute to the cellular protection from organic hydroperoxides (*Fuangthong et al., 2001*). In addition, bacillithiol has been proposed to be involved in superoxide stress and metal homeostasis, but not in hydrogen-peroxide-induced oxidative stress (*Fang and Dos Santos, 2015*; *Gaballa et al., 2010*). However, much less is known about the mechanism protecting cell-surface proteins from oxidative damages in Gram-positive bacteria. In the Gram-positive *Streptococcus pneumoniae*, surface-exposed proteins with methionine sulfoxide residues are reduced by the membrane-bound methionine sulfoxide

reductase MsrAB2 (*Saleh et al., 2013*), but the gene of an equivalent membrane-anchored enzyme is absent from the *B. subtilis* genome. To our best knowledge, YjbI is the first example of an antioxidant protein involved in protecting cell-surface proteins against oxidative aggregation. Antioxidation of cell-surface proteins is important, especially for proteins located and exposed at the air-liquid interface, such as the biofilm surface (*Beloin et al., 2004*; *Ezraty et al., 2017*). Disrupting *yjbI* had a crucial effect on the integrity of the biofilm cell surface, presumably because of this reason (*Figure 1*).

*M. tuberculosis* trHbO has an autokinase activity and plays a role in the survival and adaptation of the bacterium under hypoxia (*Hade et al., 2020*). However, *M. tuberculosis* trHbO shares a moderate (31%) amino acid sequence identity with *B. subtilis* YjbI, and *M. tuberculosis* carries another trHbN (*Couture et al., 1999*), which is missing in *B. subtilis*. Moreover, because *M. tuberculosis* is an obligate aerobic bacterium, its responses under oxygen stress may be largely different from those of the member of the facultative anaerobes *B. subtilis*. Whether the YjbI orthologues found in other bacteria, including other *B. subtilis* strains, function in the suppression of oxidative protein aggregation, biofilm formation, or oxidant resistance remains an open question for future research.

In this study, we demonstrated YjbI reduces hydroperoxide groups in BSA-OOH via its peroxidase-like activity and suppressed the spontaneous aggregation of BSA-OOH. Typical haem peroxidases, such as horseradish peroxidase and lactoperoxidase, exhibit negligible activity on protein hydroperoxides (*Davies, 2016*; *Morgan et al., 2004*; *Gebicki et al., 2002*). Haemoglobin and myoglobin can react with small peptide-sized hydroperoxides but not with large protein hydroperoxides, such as BSA-OOH (*Morgan et al., 2004*). The X-ray crystal structure of *B. subtilis* YjbI (PDB ID: 1UX8) (*Giangiacomo et al., 2005*) shows this protein has a 55 $Å^2$ surface opening (*Figure 3—figure supplement 3*), which may allow direct access of bulk solvent and large molecules to the haem active site. Therefore, the structural feature probably confers the potential to react with a protein hydroperoxide on YjbI.

An electron donor is required for the proposed protein hydroperoxide peroxidase-like reaction by YjbI. We showed that the in vitro reduction of BSA-OOH and TasA-OOH by YjbI proceeded without addition of any external electron donors (*Figure 3*). This observation implies that electrons needed for the in vitro protein hydroperoxide peroxidase-like reaction may be provided from the amino acid residues of YjbI itself. Interestingly, in the hydrogen peroxide peroxidase reaction catalysed by *M. tuberculosis* trHbO, the tyrosine residues near the haem of trHbO have been suggested to serve as electron donors (*Ouellet et al., 2007*). At least two of these tyrosine residues are conserved in trHbO orthologues (Tyr[25] and Tyr[63] in YjbI), which are located close to the haem according to the results of the crystal structure analysis (*Giangiacomo et al., 2005*). In addition, tryptophan, another electron-rich amino acid residue, is thought to contribute to electron transfer in the *M. tuberculosis* trHbO orthologue (Trp[89] in YjbI) (*Ouellet et al., 2007*). In particular, tyrosine can be stabilised by providing electrons to yield either dityrosine or a series of oxidised derivatives of dityrosine, 3,4-dihydroxyphenylalanine (DOPA), DOPA semiquinone, and DOPA quinone (*Maskos et al., 1992*; *Giulivi and Davies, 2001*). In fact, as shown in this study, the mutation in Tyr[63] affected the phenotype of *B. subtilis* (*Figure 4a*, *Table 2*) and the activity of YjbI (*Figure 4b*). Taken together, we propose that the tyrosine residues in YjbI are the most promising candidates for donating electrons for the in vitro YjbI-catalysed protein hydroperoxide peroxidase-like reaction. Nevertheless, it remains unclear whether YjbI utilises an endogenous or exogenous electron donor other than the amino acid residues of YjbI itself in vivo. Previous reports have shown that *M. tuberculosis* trHbO can receive electrons from ferrocytochrome *c* (*Ouellet et al., 2007*). Alternatively, YjbH, a DsbA-like protein (*Guddat et al., 1998*) with two active-site cysteine residues, may have a potential to donate electrons for the YjbI-catalysed reduction of protein hydroperoxides (*Fabianek et al., 2000*). However, the role of YjbH as an electron donor to YjbI is inconclusive because disrupting *yjbH* did not affect the sensitivity to AAPH (*Figure 2f*), and YjbH localises to the cytosol, unlike YjbI, to function as an effector of Spx, a central regulator of stress response (*Larsson et al., 2007*). It is also possible that physiological electron donation is operated by more than one molecule, as typical haem proteins can utilise a variety of electron-rich small or large molecules as electron donors because of their wide substrate specificity (*Galaris et al., 1989*; *Hayashi et al., 1999*; *Ouellet et al., 2007*).

The localisation of YjbI to the cell surface or the biofilm matrix is consistent with its role in protecting TasA and other cell-surface proteins from ROS-induced aggregation (*Figure 5a and b*). However, there is no apparent targeting signal sequence in the amino acid sequence of YjbI, and bioinformatic analysis using SignalP (http://www.cbs.dtu.dk/services/SignalP/abstract.php) or PSORT (https://www.

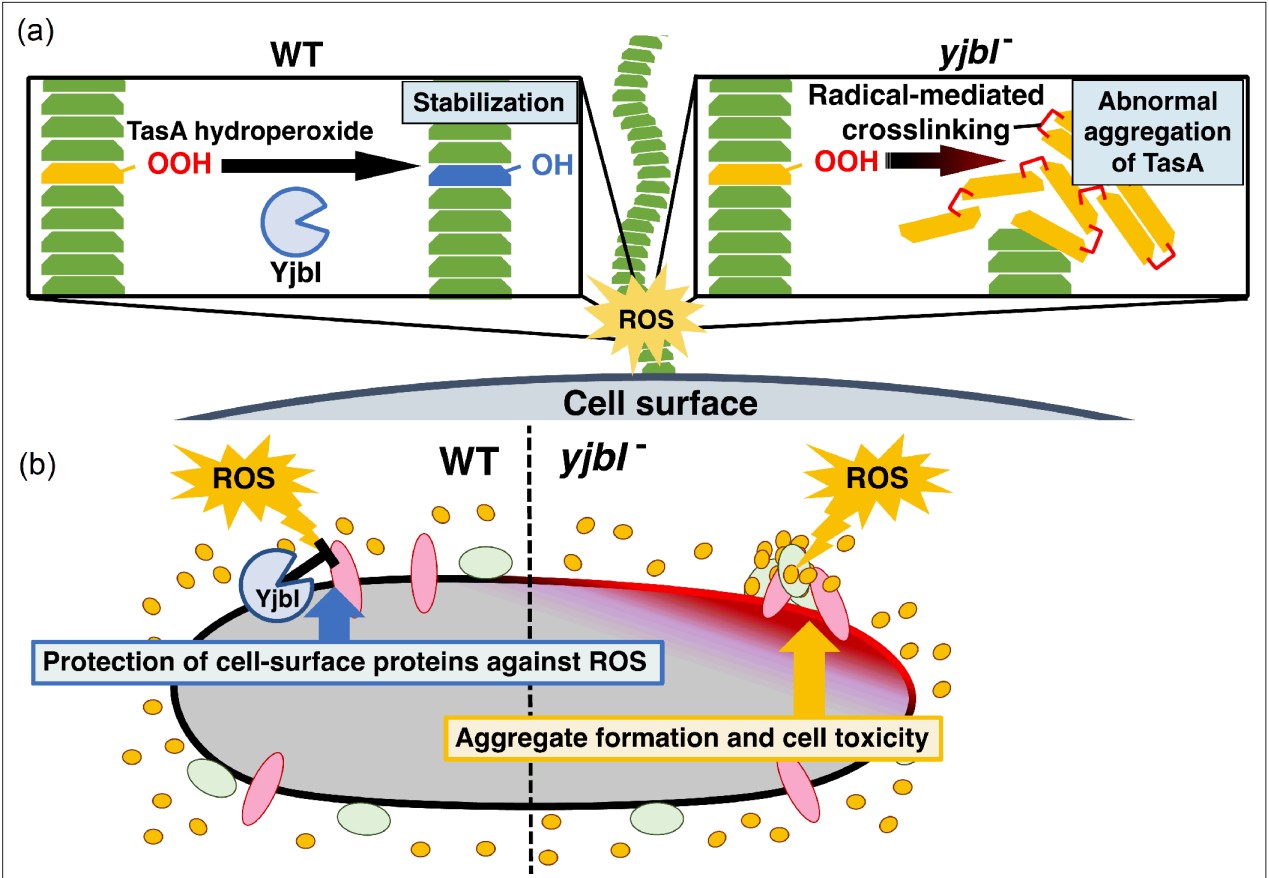

**Figure 5.** YjbI protects proteins and the cell from reactive oxygen species (ROS) by removing hydroperoxide groups from proteins. (**a**) YjbI repairs the oxidatively damaged TasA-OOH in biofilms. In the wild-type (WT) strain (left panel), the normal fibrous TasA (green trapezoids) is damaged by ROS to generate TasA-OOH (yellow trapezoid) on the cell surface. YjbI converts TasA-OOH to TasA-OH (blue trapezoid) to stabilise the normal TasA fibre. In contrast, TasA-OOH aggregates via radical-mediated protein cross-linking in the *yjbI-* strain (right panel). (**b**) A proposed role of YjbI in the general protection of cell-surface proteins from ROS-induced oxidative damage. Protein hydroperoxide-modified residues generated in various cellular surface proteins (red, green, and yellow ellipses) are generally reduced to protein hydroxy residues by YjbI to protect from further damage (i.e., aggregation) (left half). *yjbI* disruption accumulates protein aggregates via hydroperoxidation and cross-linking of various proteins, resulting in cell toxicity (right half).

The online version of this article includes the following figure supplement(s) for figure 5:

**Figure supplement 1.** *Bacillus subtilis* YjbI does not contain a predicted signal sequence.

psort.org) predicted that YjbI is a cytoplasmic protein (*Figure 5—figure supplement 1*). Purified recombinant YjbI, which was produced in *E. coli*, was obtained in a soluble form (*Figure 2—figure supplement 1*), in contrast to the insoluble feature of the native YjbI in *B. subtilis*. Analogously, *M. tuberculosis* trHbN belonging to another group of trHb family is also devoid of a targeting signal sequence, is localised to the cell surface following glycosylation, whereas the corresponding recombinant protein expressed in *E. coli* is detected in a soluble form (*Arya et al., 2013*). Nevertheless, no glycosylation site is found on YjbI. Therefore, the mechanism for the translocation of YjbI to the extracellular surface or the biofilm matrix should be investigated in future studies.

Although protein hydroperoxides are one of the major products formed because of ROS-mediated protein oxidation (*Davies et al., 1995*; *Gebicki, 1997*), (*Gebicki and Gebicki, 1999*), little is known about the effects of their generation on bacterial physiology. Biofilm matrix proteins are expected to be exposed to an air-liquid interface, which is a region with a relatively high risk of protein oxidation. Therefore, repairing the oxidatively damaged proteins on the cell surface may be important for bacterial adaptation in oxidative environments. The loss of biofilm integrity associated with the irreversible aggregation of the biofilm matrix protein TasA in the *yjbI*-deficient strain may provide an important clue to further understand the significance of protein hydroperoxide generation on the cell surface.

The findings of this study may lead to future applications. An increasing number of studies have suggested that biofilms are associated with bacterial infections in humans and with natural resistance of pathogenic bacteria to antibiotics (*Hall and Mah, 2017*). We found that *B. subtilis* biofilm formation can be suppressed by an antiserum against YjbI (*Figure 2e*) and that YjbI deficiency severely impairs the bacterial tolerance to the oxidative stress induced by AAPH (*Figure 2d*) or HClO (*Table 1*). HClO oxidises various amino acid residues such as methionine and lysine. In addition, the reaction of HClO with $O_2$ helps to form hydroxy radicals (*Candeias et al., 1993*; *Candeias et al., 1994*), which initiates protein peroxidation (*Stadtman and Levine, 2003*). Intriguingly, leukocytes employ ROS, including peroxides and HClO, in the respiratory burst during the innate immune response to bacterial infection (*Babior, 1984*). Therefore, the YjbI orthologues in pathogenic bacteria would be potential novel drug targets to inhibit pathogenic bacterial growth and biofilm formation. In fact, trHbO is implicated in *Mycobacterium* pathogenicity (*Wittenberg et al., 2002*). Future studies on the structure-function relationship of YjbI may contribute to developing a specific inhibitor that suppresses infections by the wide variety of trHbO-harbouring pathogenic bacteria (e.g., *Bacillus*, *Mycobacterium*, and *Staphylococcus*).

## Methods

### Bacterial strains, plasmids, and culture conditions

The bacterial strains and plasmids used in this study are listed in . *B. subtilis* 168 and *E. coli* derivatives were obtained by transformation of competent cells with plasmids according to standard protocols (*Harwood and Cutting, 1990*; *Studier and Moffatt, 1986*). For pre-cultivation, *B. subtilis* strains were grown in LB medium at 37°C for 8 hr. Recombinant *E. coli* strains were cultivated in LB medium at 37°C. When appropriate, 30 μg/mL tetracycline and 100 μg/mL kanamycin were added to the cultures of *B. subtilis* and *E. coli*, respectively.

### Pellicles and colony biofilm formation

Pellicles in liquid media were prepared by inoculating 1/20 volume of the *B. subtilis* pre-culture in the biofilm-promoting MSgg medium (*Bucher et al., 2016*) (5 mM potassium phosphate in 100 mM MOPS buffer at pH 7.0 supplemented with 2 mM $MgCl_2$, 700 μM $CaCl_2$, 50 μM $MnCl_2$, 50 μM $FeCl_3$, 1 μM $ZnCl_2$, 2 μM thiamine, 0.5% (w/w) glycerol, 0.5% (w/w) glutamate, 0.005% (w/w) phenylalanine, 0.005% (w/w) tryptophan, and 0.005% (w/w) threonine), followed by cultivation at 37°C for 48 hr without shaking. Bacterial colony biofilms on a solid medium were obtained by inoculating a 3 μL aliquot of the *B. subtilis* pre-culture on solid MSgg medium containing 1.5% (w/w) agar, followed by cultivation at 37°C for 48 hr.

### Biofilm surface repellence analysis

Biofilm surface repellence was evaluated according to a described method (*Arnaouteli et al., 2017*) by measuring the contact angle of a 2 μL water droplet placed on the centre of a bacterial colony biofilm formed on solid MSgg medium, using a DSA100 drop shape analyser (KRÜSS). The water droplet was equilibrated at 28°C for 5 min before imaging and measurement. The contact angle is presented as the mean ± SEM of at least three independent experiments.

### Fractionation of soluble and insoluble *B. subtilis* biofilm proteins

*B. subtilis* pellicle (0.1 g) was harvested using a spatula and lysed by incubation with 0.5 mL of B-PER (Thermo Fisher Scientific) saturated with phenylmethylsulfonyl fluoride (PMSF, Sigma-Aldrich), supplemented with 5 mg/mL lysozyme at 28°C for 2 hr, and centrifuged. Both the soluble proteins in the supernatant and the precipitated insoluble proteins were recovered and used for further studies.

### Purification of YjbI and TasA produced in *E. coli*

To purify recombinant YjbI, *E. coli* BL21(DE3)/pEyjbI2 cells were grown in LB medium at 37°C for 6 hr, followed by induction of gene expression with 25 μM isopropyl β-ᴅ-1-thiogalactopyranoside and further cultivation for 10 hr. After centrifugation, harvested cells were lysed in B-PER (Thermo Fisher Scientific) supplemented with approximately 1 mM PMSF according to the manufacturer's instructions. After centrifugation, the crude extract was fractionated with ammonium sulphate (30–60%

saturation) for 16 hr at 4°C. After centrifugation, the protein precipitate was resuspended in 50 mM MOPS buffer (pH 7.0) and then applied to a Sephacryl S-100 column (GE Healthcare) equilibrated with the same buffer. Red haemoprotein fractions from the column were collected and passed through an Amicon Ultra 100 kDa device (Millipore). The filtrate containing YjbI was applied to a Toyopearl DEAE-650M column (Tosoh) equilibrated with 50 mM Tris-acetate buffer (pH 7.4). YjbI was eluted at a flow rate of 0.25 mL/min with a linear NaCl gradient (0–0.5 M) prepared in the same buffer. The fractions containing YjbI were desalted and concentrated using an Amicon Ultra 10 kDa device in 50 mM Tris-acetate buffer (pH 7.4) to yield the purified YjbI preparation.

His-SUMO-YjbI(Y63F) and His-SUMO-TasA$_{28-261}$ were produced in *E. coli* BL21(DE3)/pEyjbI3 and *E. coli* BL21(DE3)/pEtasA3, respectively and cultured in LB medium at 37°C for 6 hr, followed by induction with 25 μM isopropyl β-ᴅ-1-thiogalactopyranoside and further cultivation for 10 hr. Cells were harvested by centrifugation and lysed with B-PER (Thermo Fisher Scientific) supplemented with approximately 1 mM PMSF. The crude extract was applied to an immobilised metal affinity chromatography (IMAC) resin charged with cobalt (HisTALON Gravity Columns; Takara Bio) and the target proteins were eluted according to the manufacturer's instructions. The desalted His-SUMO-YjbI(Y63F) and His-SUMO-TasA$_{28-261}$ were then treated with SUMO protease (SUMO protease; Sigma-Aldrich) and passed through the column again to obtain untagged Y63F and TasA$_{28-261}$ derivatives from the wash fraction, respectively.

TasA$_{28-261}$-His, a mature form of TasA (*Diehl et al., 2018*) (amino acid residues 28–261) with a His-tag at the C-terminus was produced in *E. coli* BL21(DE3)/pEtasA2 by culturing the cells in LB medium at 37°C for 6 hr, followed by induction with 25 μM isopropyl β-ᴅ-1-thiogalactopyranoside and further cultivation for 10 hr. The cells were harvested by centrifugation and lysed in B-PER (Thermo Fisher Scientific) supplemented with approximately 1 mM PMSF. The crude extract was applied onto a nickel chelation column (His GraviTrap, GE Healthcare), and TasA$_{28-261}$-His was eluted according to the manufacturer's instructions.

## In vitro ROS-induced TasA aggregation

Purified TasA$_{28-261}$-His (200 μg) was incubated with 1 mM hydrogen peroxide and 10 μM FeCl$_2$ in the presence or absence of YjbI (15 μg) in 50 mM Tris-acetate buffer (pH 7.4) at 37°C for 30 min. The proteins, after being precipitated by adding 4 volumes of cold acetone, were harvested by centrifugation, and dissolved in a standard SDS sample buffer (0.125 M Tris (pH 6.8), 10% (w/w) 2-mercaptoethanol, 4% (w/w) SDS, 10% (w/w) sucrose, 0.01% (w/w) bromophenol blue), which was followed by SDS-PAGE and western blot analysis of TasA$_{28-261}$-His with an anti–His-tag antibody.

## Analysis of YjbI cell-surface localisation following extracellular protease digestion

YjbI cell-surface localisation was analysed following extracellular protease digestion of an intact *B. subtilis* pellicle. *B. subtilis* pellicle (10 mg) was incubated in 50 mM Tris-acetate buffer (pH 7.4) containing 5 mg/mL protease mixture (Pronase E, Sigma-Aldrich) in a total volume of 1 mL at 28°C for 2 hr and then directly resuspended in the standard SDS sample buffer before SDS-PAGE and western blot analysis.

## SDS-PAGE and western blot analysis

Lysates in the standard SDS sample buffer were heated and then resolved on a 4–20% SDS/PAGE gradient gel (Mini-PROTEAN, Bio-Rad), followed by electroblotting on a PVDF membrane via a Trans Blot Turbo (Bio-Rad) at 25 V for 30 min. The membrane was blocked with 1% (w/w) skim milk in TBS (50 mM Tris-HCl and 0.85% (w/w) NaCl, pH 7.4) at 37°C for 1 hr, incubated with an anti-YjbI or anti-His-tag antibody at a dilution of 1: 1000 (vol/vol) in 37°C TBS for 2 hr, washed in TBST (TBS with 0.1% (v/v) Tween-20), and then incubated with a secondary goat anti-rabbit (for the anti-YjbI antiserum) or goat anti-mouse (for the anti-His-tag antibody) antibody conjugated with horseradish peroxidase (Bio-Rad) for 2 hr. After the membrane was washed in TBST, the immunoreactive bands were visualised using 1-Step Ultra TMB (Pierce). The membrane was photographed using a gel imager (Gel Doc, Bio-Rad).

The anti-His-tag monoclonal antibody was purchased from Medical and Biological Laboratories (Nagoya, Japan), and the anti-YjbI antiserum was a rabbit serum purchased from Sigma Genosys.

The anti-YjbI antiserum was prepared by immunising rabbits with purified YjbI mixed with complete Freund's adjuvant, boosting five times with each a week apart.

## Planktonically grown *B. subtilis* strains under AAPH-induced oxidative stress

After growth in LB medium at 37°C for 8 hr, a 3 µL aliquot of the *B. subtilis* culture was inoculated (at $OD_{600}$=0.02) into 5 mL LB liquid medium containing 0, 25, 50, or 100 mM AAPH at 37°C for 31 hr. A small amount of each culture was fractionated at 0, 2, 4, 6, 8, 10, 12, 15, 20, 25, and 31 hr. Next, each fractionated culture was diluted with 0.9% (w/w) saline to keep the absorbance ≤1.0 and then measured at 600 nm.

## Minimum bactericidal concentration assay for hypochlorous acid

Pre-cultured *B. subtilis* strains (50 µL) were mixed with 950 µL of phosphate-buffered saline (137 mM NaCl, 8.1 mM $Na_2HPO_4$, 2.68 mM KCl, and 1.47 mM $KH_2PO_4$, pH 7.4) containing HClO at final concentrations between 0 and 500 mM. After incubation at 37°C for 1 hr, a 10 µL aliquot from each mixture was spread on LB solid medium. The cells were grown at 37°C for 48 hr before bacterial colony counting. Then, the minimum concentration of HClO that generated no colony was determined. Tetracycline was not added to the pre-culture and evaluation solid media in the *yjbI⁻/yjbIʷᵀ* and point-mutated strains to ensure that tetracycline did not cause oxidative damage synergistically with HClO.

## Generation of protein hydroperoxides

BSA-OOH was generated by the Fenton reaction or oxidation with AAPH (*Gieseg et al., 2000*). For the Fenton reaction, 0.2 mg/mL BSA (fatty-acid-free, Sigma-Aldrich) was incubated with 150 mM hydrogen peroxide and 1 mM $FeCl_2$ in 150 mM acetate buffer (pH 5.0) in a total volume of 30 µL at 37°C for 15 min. To remove excess hydrogen peroxide and iron, 1 mL of 80% cold acetone was added and the sample was centrifuged at 20,000× *g* for 10 min to precipitate BSA-OOH. For oxidation with AAPH, 0.2 mg/mL BSA or TasA was incubated with 25 mM AAPH in 50 mM Tris-acetate buffer (pH 7.0) in a total volume of 30 µL at 37°C for 12 hr. The reaction was terminated by adding 4 volumes of cold acetone. After centrifugation, precipitated BSA-OOH or TasA-OOH was recovered.

## Analysis of the *YjbI*-mediated prevention of ROS-induced protein aggregation

The effect of YjbI on ROS-induced BSA aggregation was examined in two experiments: a two-step reaction and a one-pot reaction. In the two-step reaction, BSA-OOH was prepared beforehand as described above and then incubated with 0.1 mg/mL YjbI in 50 mM Tris-acetate buffer (pH 7.0) at 37°C for 3 hr in a total volume of 15 µL. In the one-pot reaction, each 0.1 mg/mL globins (YjbI, Hb, or Mb) and 25 mM AAPH was simultaneously added to 0.2 mg/mL BSA in 50 mM Tris-acetate buffer (pH 7.0) in a total volume of 30 µL and incubated at 37°C for 3 hr. For both experiments, the reaction was stopped by adding 4 volumes of ice-cold acetone, and the precipitated proteins were recovered by centrifugation. Proteins were resolved by SDS-PAGE and visualised by silver staining (Wako, Silver Stain II Kit).

## Determination of the hydroperoxide groups in BSA

After hydroperoxidation, the acetone-precipitated BSA was dissolved in 25 mM $H_2SO_4$ to assess for the hydroperoxide groups on the protein, as described (*Gieseg et al., 2000*). In brief, 9 volumes of dissolved BSA-OOH or TasA-OOH were mixed with 1 volume of 5 mM ferrous ammonium sulphate and 5 mM xylenol orange in 25 mM $H_2SO_4$ and incubated at 28°C for 30 min in the dark. The $A_{560}$ of the mixture was measured using a UV-Vis spectrophotometer (SH-9000, Corona Electric). The molar extinction coefficient of the xylenol orange-$Fe^{3+}$ complex ($1.5 \times 10^4$ $M^{-1}cm^{-1}$) (*Gieseg et al., 2000*) in 25 mM $H_2SO_4$ at 28°C was used to determine the hydroperoxide concentration. The number of hydroperoxide residues per BSA subunit or mature TasA molecule (amino acids 28–261) was determined by calculating the mole of peroxide generated per mole of added BSA (molecular weight: 66,463) or mature TasA (molecular weight: 25,716).

## Statistical analyses

Statistical differences between two experimental groups were identified using one-tailed Student's *t*-test assuming equal variance (Microsoft Excel for Mac v.16.16.21). No data points were removed from the data set before the analyses.

## Strains and plasmids used in this study

| Strain | Genotype | Source |
|---|---|---|
| *Bacillus subtilis* 168 | *trpC2* | |
| BFS2846 | *B. subtilis* 168 *yjbI::erm* | NBRP*[1] |
| COTNd | *B. subtilis* 168 *tasA::erm* | NBRP*[1] |
| BKE11550 | *B. subtilis* 168 *yjbH::erm* | NBRP*[1] |
| TKSYcomp | pHyjbI1/BKE11550 | This work |
| TKSY25 | pHyjbI4/BKE11550 | This work |
| TKSY63 | pHyjbI5/BKE11550 | This work |
| TKSY89 | pHyjbI6/BKE11550 | This work |

| Plasmid | Description | Source |
|---|---|---|
| pHY300PLK | A vector with *hsrM1*, *leuA8*, *metB5*, and *tet* for stable gene expression in *B. subtilis* | TaKaRa Bio |
| pET28b(+) | An overexpression vector with a T7 promoter and *kan* | Novagen |
| pHyjbI1 | A synthetic *yjbI* gene fragment*[2], whose sequence is shown below, was inserted into the EcoRI-BamHI sites of pHY300PLK: GAATTCATGGGACAATCCTTTAACGCCCCTTATGAAGCAATCGGTGAAGAACTTTTATCTCAGTTGGTTGACACATTTTACGAACGGGTAGCGTCACATCCGCTGCTTAAACCGATTTTTCCGAGCGATCTGACAGAAACAGCAAGAAACAAAAACAGTTTCTGACACAATATCTGGGCGGACCGCCGCTGTATACAGAAGAACATGGCCATCCGATGCTTAGAGCTCGCCATTTACCGTTTCCGATTACAAATGAACGCGCAGATGCGTGGCTGTCTTGCATGAAAGATGCCATGGATCATGTTGGCCTGGAAGGAGAAATCAGAGAATTTCTTTTTGGACGCCTGGAACTGACAGCAAGACATATGGTGAATCAAACGGAAGCCGAAGACAGAAGCAGCTAAGGATCC | This work |
| pHyjbI4 | A synthetic Y25F *yjbI* gene fragment*[2], whose sequence is shown below, was inserted into the EcoRI-BamHI sites of pHY300PLK: GAATTCATGGGACAATCGTTTAACGCACCTTATGAAGCGATTGGAGAGGAACTTCTATCGCAACTTGTTGATACTTTTTATGAGCGTGTCGCGTCTCATCCTTTGCTGAAGCCGATTTTTCCAAGCGATTTGACAGAAACCGCCAGGAAACAGAAGCAATTCTTAACTCAGTATTTAGGCGGGCCTCCTCTTTATACTGAGGAACACGGCCATCCTATGCTCAGAGCAAGGCATCTTCCCTTTCCAATTACAAACGAGAGAGCTGATGCGTTTCTCAGCTGTATGAAGGACGCAATGGACCATGTAGGGCTGGAGGGCGAAATTCGTGAGTTTTTGTTTGGCCGGCTGGAGTTGACAGCAAGGCATATGGTGAATCAAACGGAAGCGGAGGATCGATCATCTTGAGGATCC | This work |

*Continued on next page*

*Continued*

| Strain | Genotype | Source |
|---|---|---|
| pHyjbI5 | A synthetic Y63F *yjbI* gene fragment*[2], whose sequence is shown below, was inserted into the EcoRI-BamHI sites of pHY300PLK:<br>**GAATTC**<u>ATG</u>GGACAATCGTTTAACGCACCTTAT GAAGCGATTGGAG AGGAACTTCTATCGCAACTTGTTGATACTTTTTATG AGCGTGTCGCG TCTCATCCTTTGCTGAAGCCGATTTTTCCAAGCGAT TTGACAGAAAC CGCCAGGAAACAGAAGCAATTCTTAACTCAGT ATTTAGGCGGGCCT CCTCTTTTTACTGAGGAACACGGCCATCCTATGCTC AGAGCAAGGC ATCTTCCCTTTCCAATTACAAACGAGAGAGCTGATG CGTGGCTCAG CTGTATGAAGGACGCAATGGACCATGTAGGGC TGGAGGGCGAAATT CGTGAGTTTTTGTTTGGCCGGCTGGAGTTGAC AGCAAGGCATATGG TGAATCAAACGGAAGCGGAGGATCGATCATCT <u>TGA</u>**GGATCC** | This work |
| pHyjbI6 | A synthetic W89F *yjbI* gene fragment*[2], whose sequence is shown below, was inserted into the EcoRI-BamHI sites of pHY300PLK:<br>**GAATTC**<u>ATG</u>GGACAATCGTTTAACGCACCTTAT GAAGCGATTGGAG AGGAACTTCTATCGCAACTTGTTGATACTTTTTATG AGCGTGTCGCG TCTCATCCTTTGCTGAAGCCGATTTTTCCAAGCGAT TTGACAGAAAC CGCCAGGAAACAGAAGCAATTCTTAACTCAGT ATTTAGGCGGGCCT CCTCTTTATACTGAGGAACACGGCCATCCTATGCTC AGAGCAAGGC ATCTTCCCTTTCCAATTACAAACGAGAGAGCTGATG CGTTTCTCAGC TGTATGAAGGACGCAATGGACCATGTAGGGCT GGAGGGCGAAATTC GTGAGTTTTTGTTTGGCCGGCTGGAGTTGACA GCAAGGCATATGGT GAATCAAACGGAAGCGGAGGATCGATCATCT<u>TGA</u> **GGTACC** | This work |
| pHtasA1 | A synthetic *tasA* gene fragment*[2], whose sequence is shown below, was inserted into the EcoRI-BamHI site of pHY300PLK to produce TasA with a C-terminal His$_6$-tag (TasA-His):<br>**GAATTC**<u>ATG</u>GGGAATGAAGAAAAAATTAAGCCTG GGAGTGGCATCTGCAGCTTTAGGATTAGCGC TGGTCGGCGGAGGCACATGGGCTGCATTTAAC GATATCAAATCTAAAGATGCAACATTTGCGTCA GGCACACTTGATTTAAGCGCTAAAGAAAACTC AGCCAGCGTCAATTTAAGCAACCTGAAACCGG GAGATAAACTGACAAAAGATTTTCAATTTGAAAATA ACGGCTCTCTGGCTATCAAAGAAGTTCTT ATGGCCCTGAACTACGGAGATTTTAAAGCAAA CGGAGGCTCTAACACATCACCGGAAGATTTTCT GTCACAGTTTGAAGTCACACTGCTTACAGTTG GAAAAGAAGGCGGCAATGGCTATCCGAAAAAC ATTATCCTGGATGATGCCAATCTTAAAGATCTGTAC CTTATGTCAGCAAAAAACGATGCAGCGGCT GCCGAAAAAATCAAAAAACAAATCGATCCGAA ATTTCTGAACGCAAGCGGCAAAGTTAACGTGG CGACAATTGATGGAAAAACAGCTCCGGAATAT GATGGCGTTCCGAAAACACCGACAGATTTTGAT CAAGTGCAGATGGAAATCCAGTTTAAAGATGA TAAAACAAAAGATGAAAAAGGACTTATGGTGC AAAACAAATACCAGGGCAACTCTATCAAACTTCAAT TTTCATTTGAAGCGACACAGTGGAATGG ACTGACAATTAAGAAAGATCATACAGATAAAG ATGGCTACGTTAAAGAAAACGAAAAAGCTCAT TCAGAAGATAAAAACCATCATCATCATCATCAT<u>TAA</u> **GGATCC** | This work |

*Continued on next page*

*Continued*

| Strain | Genotype | Source |
|--------|----------|--------|
| | A synthetic *yjbI* gene fragment*[2], whose sequence is shown below, was inserted into the NcoI-BamHI site of pET28b(+): CC<u>ATG</u>GGTCAGAGCTTTAACGCGCCGTACGAGGCGATCGGCGAGGAACTGCTGAGCCAACTGGTGGACACCTTCTATGAACGTGTTGCGAGCCACCCGCTGCTGAAGCCGATTTTTCCGAGCGATCTGACCGAAACCGCGCGTAAGCAGAAACAATTCCTGACCCAGTACCTGGGTGGCCCGCCGCTGTATACCGAGGAACACGGTCACCCGATGCTGCGTGCGCGTCACCTGCCGTTCCCGATCACCAACGAACGTGCGGACGCGTGGCTGAGCTGCATGAAAGACGCGATGGATCACGTGGGTCTGGAGGGCGAAATTCGTGAGTTCCTGTTTGGCCGTCTGGAACTGACCGCGCGTCACATGGTTAACCAAACCGAGGCGGAAGATCGTAGCAGC<u>TAA</u> | |
| pEyjbI2 | **GGATCC** | This work |

*[1]NBRP: National BioResource Project, Japan.
*[2]DNA fragments were synthesised with codon optimisation by GenScript. Bold letters indicate restriction enzyme sites. Initiation and termination codons are underlined.

# Acknowledgements

The authors thank K Kitayama and T Kurihara for help with amino acid sequence analysis. The gene-disrupted strains used in this research were provided by the National BioResource Project (NBRP), Japan. This work was supported by JSPS KAKENHI grant numbers JP18K14383 and JP20K15446, by the Program for the Third-Phase R-GIRO Research from Ritsumeikan University, and by a grant from the Japan Foundation for Applied Enzymology.

# Additional information

## Funding

| Funder | Grant reference number | Author |
|--------|------------------------|--------|
| Japan Society for the Promotion of Science | 18K14383 | Takeshi Imai |
| Japan Society for the Promotion of Science | 20K15446 | Takeshi Imai |
| Ritsumeikan Global Innovation Research Organization, Ritsumeikan University | | Hisaaki Mihara |

The funders had no role in study design, data collection and interpretation, or the decision to submit the work for publication.

## Author contributions

Takeshi Imai, Conceptualization, Resources, Formal analysis, Funding acquisition, Validation, Investigation, Visualization, Methodology, Writing – original draft, Project administration, conceived the idea and performed most experiments. T.I; Ryuta Tobe, Resources, Data curation, Methodology; Koji Honda, Formal analysis, Methodology; Mai Tanaka, Formal analysis, Validation; Jun Kawamoto, Supervision, Methodology; Hisaaki Mihara, Resources, Supervision, Funding acquisition, Investigation, Methodology, Project administration, Writing - review and editing

## Author ORCIDs
Takeshi Imai http://orcid.org/0000-0001-7387-6212
Hisaaki Mihara http://orcid.org/0000-0002-5153-4403

## Decision letter and Author response
Decision letter https://doi.org/10.7554/eLife.70467.sa1

Author response https://doi.org/10.7554/eLife.70467.sa2

## Additional files

### Supplementary files
• Transparent reporting form

### Data availability
All data is available within the text, figures, and tables of the manuscript. Source data files have been provided for Figures 2, 3, Figure 2-figure supplement 1, Figure 3-figure supplement 1 and Figure 4-figure supplement 1.

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
