## [Editor Report]

This manuscript is of interest to microbiologists interested in biofilm formation, haem proteins, and cellular responses to extracellular oxidative stress. It proposes a mechanism for biofilm protection from reactive oxygen species (ROIs) through the examination of the Gram positive, *Bacillus subtilis*. The results highlight the importance of protein peroxidase activity for proper biofilm assembly in a model species.

---

## [Decision Letter]

**Decision letter after peer review:**

Thank you for submitting your article "Group II truncated haemoglobin YjbI prevents reactive oxygen species-induced protein aggregation in *Bacillus subtilis*" for consideration by *eLife*. Your article has been reviewed by 2 peer reviewers, and the evaluation has been overseen by a Reviewing Editor and Carlos Isales as the Senior Editor. The following individuals involved in review of your submission have agreed to reveal their identity: Matthew Cabeen (Reviewer #2).

Essential revisions:

(1) Provide a direct experimental evidence for the proposed link between TasA and Yjbl (for example with titration NMR studies suggested by referee 1 point 4).

(2) Provide further evidence for the connection between the peroxidase activity of Yibl and its mutant phenotype in the biofilm (for example developing the mutant strain suggested by referee 2).

(3) Measure stiffness using the AFM in Force mode (referee 1 point 2) or head to head high resolution AFM comparison (referee 2 point 2).

(4) discuss why the WT and tasA colonies are more flat than in the reference paper, or why this point is not relevant for this study.

(5) Provide evidence or remove the connection between cross linking and lack of monomeric TasA in the mutant (referee 1 point 3).

(6) Clarify why you expect stiffness and wettability to be connected (in referee 2 point 2).

Please also consider all other points raised in the reviews and address them as you see fit.

*Reviewer #1 (Recommendations for the authors):*

1. The term 'disruptant' is rather unclear. I suggest to use 'mutant' instead.

2. I am not quite sure that the term 'elusive' is appropriate where it was used.

3. Figure 2. Quality of the gel is low.

4. Fenton reaction is mentioned in passing in the text. I suggest to elaborate in a sentence or two on what it is and how it is relevant to the study, for better understanding of its relevance by a general audience reader.

5. I suggest to move figure S2 to the main text in order to explain the mechanism better.

*Reviewer #2 (Recommendations for the authors):*

There are a few places where the results seem overinterpreted or are insufficiently explained. The claim (l. 331) that YjbI may be crucial for bacterial survival is not supported, as mutants have no growth defect except under >25mM AAPH treatment. The claim that YjbI has "biologically unprecedented" protein hydroperoxide peroxidase activity warrants more explanation. Is this new for a haem protein, or new for any enzyme? Is YjbI markedly better than other molecules like selenomethonine or enzymes like peroxiredoxins at removing protein hydroperoxides? More explanation would help the reader appreciate the novelty and importance of this finding.

Another way to strengthen this work is to broaden its applicability. *B. subtilis* is an excellent model organism for studying biofilm formation, but many other biofilm-forming bacterial species use different matrix components that do not include amyloid-like proteins. Showing biofilm or survival phenotypes in a few other species (and in *B. subtilis* in other conditions) when a YjbI homolog is inactivated would be evidence that such peroxidase activity is generally important and not a niche phenomenon in this one species under biofilm-forming conditions.

[Editors’ note: further revisions were suggested prior to acceptance, as described below.]

Thank you for resubmitting your work entitled "Group II truncated haemoglobin YjbI prevents reactive oxygen species-induced protein aggregation in *Bacillus subtilis*" for further consideration by *eLife*. Your revised article has been evaluated by Carlos Isales (Senior Editor) and a Reviewing Editor.

The manuscript has been improved but there are some remaining issues that need to be addressed, as outlined below:

Please consider the recommendations of the reviewers either by editing the text or by performing the experiments suggested. Note that we will accept the manuscript even without additional experiments. However, we believe some of the experiments suggested should be relatively straightforward and strengthen considerably the results.

In particular, we strongly encourage you to perform HOCl sensitivity tests and/or AAPH growth curves with the three new mutants, i.e. point (1) of Reviewer #2. If you have expertise on point (2) of Reviewer #2 please consider doing these experiments, or alternatively comment on the issue.

*Reviewer #1 (Recommendations for the authors):*

The authors have answered all my questions and addressed my concerns. This manuscript is now greatly improved!

One note: Please change gram-positive to Gram-positive throughout the manuscript.

Reviewer #2 (Recommendations for the authors):

This revised manuscript by Imai et al. describes the role of the *B. subtilis* truncated Hb protein YjbI in reversing protein hydroperoxidation, hence reducing protein aggregation and abetting normal biofilm formation. YjbI also appears to generally protect cells from ROS even in planktonic growth, and Y63F substitution in the protein abrogates both its in vitro ability to reverse protein hydroperoxidation and its role in normal biofilm wettability. The authors deserve applause for their careful responses and additional experimentation to successfully address reviewer concerns about the original manuscript. The revised manuscript is stronger in its link between physiological phenotypes (biofilm formation, ROS sensitivity) and YjbI activity and reads well. I hesitate to suggest additional experimentation, but a few quite simple experiments could further strengthen the conclusions of the paper and/or broaden its significance.

(1) The correspondence between the increased wettability of the Y63F (but not Y25F or W89F) mutant (Figure 1b) and its inability to remove hydroperoxide groups from BSA (Figure 3g) is good. The link between protein sequence and phenotype would be even stronger if the other point mutants were tested for activity. An easier assay than the hydroperoxidation assay would simply be to examine HOCl sensitivity, as the discrepancy between WT and the yjbI mutant is large. Does only the Y63F, but not the other point mutants, show HOCl sensitivity? AAPH growth curves would also serve the same purpose.

(2) Is there a reason to think that YjbI is somehow tethered to the cell surface rather than simply secreted? The manuscript text seems to suggest that it is surface localized, but none of the experiments formally show the localization. Either the wording should be modified, or a test for localization should be performed. One relatively straightforward test would be immunofluorescence microscopy of whole cells, grown planktonically or in a pellicle, using their anti-YjbI Ab as the primary and then a fluorescent secondary Ab. Is the fluorescence localized to the cell surface or to the biofilm matrix (e.g., to TasA)?

(3) The significance of the paper would be considerably broadened with at least one other example of YjbI importance or function in another species or even in another strain. The authors state in their rebuttal letter that such studies are underway and will be reported separately. I am sensitive to that argument, especially as some of the other teHbO-encoding species might be difficult to manipulate genetically. But what about moving the yjbI::erm mutation into strain 3610 via phage transduction, which would be trivial? Does the mutation have an even more severe effect in a robust biofilm-former, or does the relatively low biofilm-forming ability of 168 sensitize it to loss of YjbI activity?

---

## [Author Response]

Essential revisions:(1) Provide a direct experimental evidence for the proposed link between TasA and Yjbl (for example with titration NMR studies suggested by referee 1 point 4).

To provide direct evidence of the link between TasA and YjbI, hydroperoxidised TasA (TasA-OOH) was prepared using purified mature TasA and incubated with YjbI to examine the elimination of TasA-OOH by YjbI in vitro. The results showed that the amount of hydroperoxide residues per TasA was significantly decreased, as shown in Figure 3f in the revised manuscript, demonstrateing direct reaction of YjbI with hydroperoxidised TasA. Accordingly, revised Figure 3f and its corresponding description (lines 211–215) have been added to the revised manuscript:

“To provide direct experimental evidence of the proposed link between TasA and Yjbl, we purified mature TasA (Figure 2—figure supplement 1) and prepared TasA-OOH using the same procedure as for BSA-OOH. The addition of YjbI to TasA-OOH significantly decreased its hydroperoxide groups (Figure 3f), as observed in BSA-OOH.”

According to the above revisions, the following text and figures have been modified:

Deletions:

line 33 in the original manuscript: “prevented its”

line 468 in the original manuscript: “BSA”

Additions:

line 33 in the revised manuscript: “biofilm component TasA, preventing”

line 505 in the revised manuscript: “or TasA”

line 508 in the revised manuscript: “or TasA-OOH”

line 524 in the revised manuscript: “or TasA-OOH”

line 530 in the revised manuscript: “or mature TasA molecule (amino acids 28-261)”

lines 531–532 in the revised manuscript: “or mature TasA (molecular weight:25,716)”

Figure 3f and the legend to Figure 3 (lines 854–858) in the revised manuscript:

“(f) TasA-OOH (30 μM) in 50 mM Tris-acetate buffer (pH 7.0) was incubated with YjbI (3.3 μM) at 37°C, and the reaction was terminated by adding four volumes of cold acetone after incubation for 20 min. The number of hydroperoxide groups per TasA was determined for each sample. Data are shown as the mean ± SD of three independent experiments (***p < 0.005; t-test, one-tailed).”

Figure 3—figure supplement 1 and the legend (lines 889–898) in the revised manuscript:

“(a) Mature TasA was produced in *E. coli* BL21(DE3)/pEtasA3 cells grown in LB medium. Proteins from each purification step, i.e. soluble crude extracts (lane 1), elution fraction from immobilized metal affinity chromatography (IMAC) resin charged with cobalt (lane 2), and wash fraction from IMAC after treatment with SUMO-protease (lane 3), were resolved by SDS-PAGE and stained with Coomassie Brilliant Blue. (b) Y63F derivative of YjbI was produced in *E. coli* BL21(DE3)/pEyjbI3 cells grown in LB medium. Proteins from each purification step, soluble crude extracts (lane 1), elution fraction from IMAC (lane 2), and wash fraction from IMAC after treatment with SUMO-protease (lane 3) were resolved by SDS-PAGE and stained with Coomassie Brilliant Blue.”

(2) Provide further evidence for the connection between the peroxidase activity of Yibl and its mutant phenotype in the biofilm (for example developing the mutant strain suggested by referee 2).

Based on these suggestions, we prepared three variants of YjbI: Y25F, Y63F, and W69F. Each variant was introduced into a yjbI-deficient strain and tested for biofilm formation. Among the strains harboring the variants, only Y63F could not complement the defective phenotype of the yjbI-deficient strain in the biofilm repellency assay (revised Figure 1b). Furthermore, we confirmed that purified Y63F lost its hydroperoxide peroxidase activity (revised Figure 3g). These results provide evidence for the connection between the hydroperoxide peroxidase activity of YjbI and the abnormal biofilm phenotype of the yjbI-deficient strain. Accordingly, revised Figure 1b and 3g have been included in the manuscript, and the corresponding descriptions have been added:

lines 219–235 in the revised manuscript (new paragraph): “Effect of amino acid substitution in YjbI on its in vivo and in vitro functions”

lines 322–324 in the revised manuscript: “In addition, tryptophan, another electron-rich amino acid residue, is thought to contribute to electron transfer in the *M. tuberculosis* trHbO ortholog (Trp^89^ in YjbI) (Ouellet et al., 2007).”

lines 327–328 in the revised manuscript: “In fact, as shown in this study, the mutation in Tyr^63^ affected the phenotype of *B. subtilis* (Figure 1b) and the activity of YjbI (Figure 3g).”

According to the above revisions, the following text and figures have been modified:

Additions:

lines 34–36 in the revised manuscript: “Furthermore, the replacement of Tyr^63^ near the haem of YjbI with phenylalanine resulted in the loss of its protein peroxidase-like activity, and the mutant gene failed to rescue biofilm water repellency in the yjbI-deficient strain.”

lines 432–442 in the revised manuscript: “His-SUMO-YjbI(Y63F) and His-SUMO-TasA_28-261_ were produced in *E. coli* BL21(DE3)/pEyjbI3 and *E. coli* BL21(DE3)/pEtasA3, respectively (Supplementary File 1) and cultured in LB medium at 37°C for 6 h, followed by induction with 25 µM isopropyl β-d-1-thiogalactopyranoside and further cultivation for 10 h. Cells were harvested by centrifugation and lysed with B-PER (Thermo Fisher Scientific) supplemented with approximately 1 mM PMSF. The crude extract was applied to an immobilized metal affinity chromatography (IMAC) resin charged with cobalt (HisTALON Gravity Columns; Takara Bio) and the target proteins were eluted according to the manufacturer’s instructions. The desalted His-SUMO-YjbI(Y63F) and His-SUMO-TasA_28-261_ were then treated with SUMO protease (SUMO protease; Σ-Aldrich) and passed through the column again to obtain untagged Y63F and TasA_28-261_ derivatives from the wash fraction, respectively.”

Figure 1b and lines 783–786 in the revised manuscript: “(b) Gene complementation study using the yjbI-deficient strain of *B. subtilis* with plasmids expressing YjbI derivatives (Y25F, Y63F, and W89F). The morphology and surface repellence of the biofilms of the yjbI-deficient strains carrying each plasmid are shown.”

Figure 3g and lines 858–862 in the revised manuscript: “(g) BSA-OOH (30 μM) in 50 mM Tris-acetate buffer (pH 7.0) was incubated with the YjbI derivative Y63F (3.3 μM) at 37°C, and the reaction was terminated by adding four volumes of cold acetone after incubation for 20 min. The number of hydroperoxide groups per BSA subunit was determined for each sample. Data are shown as the mean ± SD of three independent experiments.”

Figure 3—figure supplement 1 and lines 894–899 in the revised manuscript: “(b) Y63F derivative of YjbI was produced in *E. coli* BL21(DE3)/pEyjbI3 cells grown in LB medium. Proteins from each purification step, soluble crude extracts (lane 1), elution fraction from IMAC (lane 2), and wash fraction from IMAC after treatment with SUMO-protease (lane 3) were resolved by SDS-PAGE and stained with Coomassie Brilliant Blue. The positions of the molecular mass marker proteins (M) are shown on the left.”

(3) Measure stiffness using the AFM in Force mode (referee 1 point 2) or head to head high resolution AFM comparison (referee 2 point 2).

Thanks to the referee’s comments, we have realized that our interpretations of the AFM data were not appropriate nor relevant to biofilm repellency. After careful consideration, we have concluded that the AFM part is not essential for this study, and therefore, must be removed to avoid unnecessary confusion.

According to the above revisions, the following text and figures have been modified:

Deletions:

lines 27–28 in the original manuscript: “and the mechanical stiffness of the cell surface”

lines 83–84 in the original manuscript: “normal stiffness, wrinkles, and”

lines 92–109 in the original manuscript: “yjbI disruption reduces the mechanical strength of *B. subtilis* biofilm. When the yjbI disruptant (BFS2846, Supplementary File 1) of *B. subtilis* 168 was cultured in a biofilm-promoting minimal (MSgg) liquid medium, we found that the pellicles, which are floating biofilms formed at the air-liquid interface, of the mutant were more fragile than those of the wild-type (WT) strain, and they were easily dispersed by scooping with a spatula (data not shown). Using atomic force microscopy (AFM), we measured the stiffness of the cell surfaces of the WT and yjbI disruptant. The cell surface of the WT bacteria from pellicles was stiffer than the surrounding matrix, as indicated in the AFM phase images by the relatively lower brightness of the cell surface (Figure 1a). Conversely, the cell surface of the yjbI disruptant pellicles showed lower stiffness than the matrix (Figure 1b), suggesting that the mutant pellicles have a less stiff cell surface than the WT. In addition, the average size of the yjbI disruptant cells (4.4 ± 0.13 µm) was significantly greater than the WT cells (3.1 ± 0.16 µm) (p < 0.001, mean ± SEM, n = 50) (Figure 1a and 1b). Introducing a plasmid encoding yjbI into the yjbI disruptant partially rescued the effect of yjbI disruption, resulting in cell-surface stiffness higher than the stiffness of the matrix and an average length of 3.8 ± 0.19 µm (Figure 1c). These results suggest yjbI is important for maintaining the cell-surface stiffness of pellicles. The decrease in the stiffness of the cell surface upon the yjbI disruption presumably elongated the cells.” in the original manuscript.

lines 130–132 in the original manuscript: “The loss of surface repellence in the yjbI disruptant biofilm suggests that its biofilm surfaces are wetter than those of WT, which is likely to be related to the lower stiffness phenotype of this mutant strain (Figure 1a-c).”

lines 373–380 in the original manuscript: “AFM analysis of biofilms. A *B. subtilis* pellicle was gently placed on a silicon substrate for surface analysis by AFM. All measurements were carried out using a scanning probe microscope (E-Sweep; Hitachi High-Tech Science Corporation, Tokyo, Japan) equipped with a microcantilever (SI-DF3, Hitachi High-Tech Science Corporation) with a spring constant of 1.5 N/m and a resonance frequency of 26 kHz. The relative surface softness and the length of the long axis of 50 individual bacterial cells in an AFM image were analysed for each strain using ImageJ (NIH, v.1.52k) (https://imagej.nih.gov/ij/).”

Line 515 in the original manuscript: “AFM”

lines 729–736 in the original manuscript: “Phase-imaging atomic force microscopy (AFM) was used to analyze the wild-type (WT) (a), yjbI-disrupted mutant (yjbI^-^) (b), and yjbI^-^ complemented with a yjbI­­-carrying plasmid (yjbI^-^/yjbI^+^) (c) of *B. subtilis* biofilm cells from the pellicles on liquid MSgg medium. The areas indicated with cyan boxes in the upper panels were magnified to produce the images in the lower panels in (a), (b), and (c). The contrast represents the relative stiffness of the objects; the objects with dark pixels are stiffer than those with bright pixels within the same image. Similar AFM results were obtained from two independent experiments.”

Figure 1a, b, c in the original manuscript

(4) discuss why the WT and tasA colonies are more flat than in the reference paper, or why this point is not relevant for this study.

The differences between the morphology of colonies in our study and that of the reference study are due to the utilization of different strains of *B. subtilis* (strain 168 versus NCBI3610). In our work, we employed *B. subtilis* strain 168 because this strain is a close representative of *B. subtilis*, as described by Zeigler et al. (J. Bacteriol., 190:6983-6995, 2008), and serves as a model organism for wider aspects of basic research, including oxidative damage responses. On the contrary, studies that focus on biofilm formation, including the reference papers, generally prefer to use the undomesticated ancestral isolate of *B. subtilis* strain NCBI3610, which forms a well-structured three-dimensional biofilm (Arnaouteli et al. Nat. Rev. Microbiol., 19:600-614, 2021; Mielich-Süss and Lopez, Environ. Microbiol., 17:555-565, 2014), as a model organism. To clarify this point, we have added the following text to the revised manuscript in lines 269–277:

“Most studies on biofilm formation in *B. subtilis* use the *B. subtilis* NCBI3610 strain as a model bacterium because of its ability to form well-structured three-dimensional biofilms (Arnaouteli et al., 2021, Mielich-Süss et al., 2014). The biofilms of the wild-type and tasA mutant strains of the *B. subtilis* 168 strain are known to be morphologically different from those of the *B. subtilis* NCBI3610 strain (Romero et al., 2010, Vlamakis et al., 2008, Erskine et al., 2018). In this study, the *B. subtilis* 168 strain was used because it is the most representative of *B. subtilis* and serves as a model organism for a wider range of research aspects (Zeigler et al., 2008) as we were not only interested in evaluating biofilm formation but also in more general aspects of oxidative damage responses in bacteria.”

(5) Provide evidence or remove the connection between cross linking and lack of monomeric TasA in the mutant (referee 1 point 3).

We acknowledge the reviewer’s comments on this point, and apologize for this inappropriate statement. We have removed the statement regarding the lack of monomeric TasA in the mutant (lines 143–146 of the original manuscript). A new sentence has been included to highlight the potential involvement of covalent bonds in TasA aggregate formation in lines 126–129 in the revised manuscript.

“No monomeric TasA was detected in the insoluble fraction of the yjbI-deficient mutant strain. An aggregate of TasA was observed under strong reducing and heat-denaturing conditions in SDS sample buffer, suggesting that covalent bonds may be involved in aggregate formation.”

(6) Clarify why you expect stiffness and wettability to be connected (in referee 2 point 2).

We thank the reviewer for their comment. As explained above (answer to Essential Revisions, point 3), the AFM data and the corresponding descriptions have been removed.

Reviewer #1 (Recommendations for the authors):1. The term 'disruptant' is rather unclear. I suggest to use 'mutant' instead.

In response to this suggestion, we have replaced the word “disruptant” with “deficient mutant” or “deficient strain”.

2. I am not quite sure that the term 'elusive' is appropriate where it was used.

In response to this suggestion, we have replaced the term “elusive” with “unclear” (lines 52 and 84).

3. Figure 2. Quality of the gel is low.

The gels used in this study were obtained from Bio-Rad (4–20% SDS-PAGE gradient gel, Mini-PROTEAN). The smeared bands were most likely due to the various states of the protein aggregates.

4. Fenton reaction is mentioned in passing in the text. I suggest to elaborate in a sentence or two on what it is and how it is relevant to the study, for better understanding of its relevance by a general audience reader.

According to your suggestion, we have added a short explanatory sentence to the revised manuscript in in lines 134–136: “The Fenton reaction produces hydroxyl radicals according to the following equation: Fe^2+^ + H2O2 → Fe^3+^ + OH^-^ + ·OH, and was used to induce ROS-mediated protein oxidation.”

5. I suggest to move figure S2 to the main text in order to explain the mechanism better.

According to the reviewer’s comment, we moved Figures S2–3a in the revised manuscript.

Relevant Figure 3 rearrangements:

Figure 3a, b → Figure 3c, d

Figure 3c → Figure 3e

Reviewer #2 (Recommendations for the authors):There are a few places where the results seem overinterpreted or are insufficiently explained. The claim (l. 331) that YjbI may be crucial for bacterial survival is not supported, as mutants have no growth defect except under >25mM AAPH treatment. The claim that YjbI has "biologically unprecedented" protein hydroperoxide peroxidase activity warrants more explanation. Is this new for a haem protein, or new for any enzyme?

We thank you for the comment. This sentence has been revised as follows:

lines 215–217:

“Taken together, these results suggest that YjbI prevents protein aggregation/fragmentation, most likely through its haem-mediated protein hydroperoxide peroxidase-like activity (Figure 3a).”

lines 360–361: “Therefore, repairing the oxidatively damaged proteins on the cell surface may be important for bacterial adaptation in oxidative environment.”

We used the words “biologically unprecedented” because YjbI is the first example of a cell-surface-localized hemoprotein exhibiting protein hydroperoxide peroxidase activity. However, to avoid confusion, we have removed these terms in the revised manuscript (lines 25, 207, and 235 in the original manuscript).

Is YjbI markedly better than other molecules like selenomethonine or enzymes like peroxiredoxins at removing protein hydroperoxides? More explanation would help the reader appreciate the novelty and importance of this finding.Another way to strengthen this work is to broaden its applicability. *B. subtilis* is an excellent model organism for studying biofilm formation, but many other biofilm-forming bacterial species use different matrix components that do not include amyloid-like proteins. Showing biofilm or survival phenotypes in a few other species (and in *B. subtilis* in other conditions) when a YjbI homolog is inactivated would be evidence that such peroxidase activity is generally important and not a niche phenomenon in this one species under biofilm-forming conditions.

We have added a description of animal peroxiredoxins and selenomethionine (with GSH or a thioredoxin system) that have been shown to scavenge protein hydroperoxides. We have also added the description of how YjbI differs from peroxiredoxin and selenomethionine.

The corresponding sentences have been added to the revised manuscript in lines 254–268:

“Peroxiredoxins have been reported to repair intracellular protein peroxidation in mammals (Peskin et al., 2010). However, YjbI is distinct from peroxiredoxins in that it is a haem protein with no significant sequence homology (<15%). The second-order rate constants (M^-1^·s ^-1^) for the reactions of mammalian peroxiredoxins 2 and 3 with BSA-OOH are 160 and 360, respectively, and have been shown to reduce protein peroxides more efficiently than GSH under physiological conditions (Peskin et al., 2010). Although direct comparison is difficult due to different experimental conditions, YjbI and peroxiredoxins are likely to have a similar catalytic rate, as both proteins can reduce BSA-OOH in the order of several mM in roughly 5 min at similar protein concentrations (Figure 3e) (Peskin et al., 2010). Interestingly, selenomethionine can catalyze the removal of hydroperoxides from proteins in the presence of GSH or a thioredoxin system (Rahmanto and Davies, 2011). However, this system, as well as peroxiredoxins, localises in the cytoplasm of cells, which is a significant difference between YjbI and these proteins. Moreover, whether bacteria utilize peroxiredoxins and the selenomethionine system to remove hydroperoxides from proteins remains unclear.”

The function of YjbI in other bacterial species is currently under investigation and will be reported in a separate study in the near future.

Other modifications

Deletions:

line 12 in the original manuscript: “GSH; glutathione”

line 112 in the original manuscript: “we also”

lines 799–801 in the original manuscript: “(b) BSA-OOH prepared by AAPH treatment was incubated in the presence (+) or absence (–) of YjbI and analyzed by SDS-PAGE and silver staining. The labels on the left or right side of the images are the same as those in (a)”

lines 813–817 in the original manuscript: “(e) YjbI suppresses AAPH-induced accumulation of BSA-OOH. BSA (6.0 mM) was incubated with 25 mM AAPH in 50 mM Tris-acetate buffer (pH 7.0) with (blue bar) or without (red bar) YjbI (6.6 mM) for 3 h at 37 °C before determining the number of hydroperoxide groups per BSA subunit. The data represent the mean ± SD of three independent experiments (**p < 0.01; t-test, one-tailed).”

Additions:

line 85 in the revised manuscript: “In this study, we”

lines 547–548 in the revised manuscript: “water contact angles”

line 781 in the revised manuscript: “YjbI is needed for normal biofilm formation.”

line 818 in the revised manuscript: “YjbI exhibits protein hydroperoxide peroxidase activity.”

Supplementary File 1: Sequence information for pHyjbI4, pHyjbI5, pHyjbI6, pEyjbI3, and pEtasA3

[Editors’ note: what follows is the authors’ response to the second round of review.]

Reviewer #1 (Recommendations for the authors):The authors have answered all my questions and addressed my concerns. This manuscript is now greatly improved!One note: Please change gram-positive to Gram-positive throughout the manuscript.

Thank you for your immense suggestion. We have corrected the typo.

Reviewer #2 (Recommendations for the authors):This revised manuscript by Imai et al. describes the role of the *B. subtilis* truncated Hb protein YjbI in reversing protein hydroperoxidation, hence reducing protein aggregation and abetting normal biofilm formation. YjbI also appears to generally protect cells from ROS even in planktonic growth, and Y63F substitution in the protein abrogates both its in vitro ability to reverse protein hydroperoxidation and its role in normal biofilm wettability. The authors deserve applause for their careful responses and additional experimentation to successfully address reviewer concerns about the original manuscript. The revised manuscript is stronger in its link between physiological phenotypes (biofilm formation, ROS sensitivity) and YjbI activity and reads well. I hesitate to suggest additional experimentation, but a few quite simple experiments could further strengthen the conclusions of the paper and/or broaden its significance.(1) The correspondence between the increased wettability of the Y63F (but not Y25F or W89F) mutant (Figure 1b) and its inability to remove hydroperoxide groups from BSA (Figure 3g) is good. The link between protein sequence and phenotype would be even stronger if the other point mutants were tested for activity. An easier assay than the hydroperoxidation assay would simply be to examine HOCl sensitivity, as the discrepancy between WT and the yjbI mutant is large. Does only the Y63F, but not the other point mutants, show HOCl sensitivity? AAPH growth curves would also serve the same purpose.

Following your advice, HClO sensitivity experiments were carried out using the complementary strains of each yjbI mutant (Y25F, Y63F, or W89F). The results are shown in the revised Table 2, where the HClO resistance was partially restored in the yjbI(WT)complementary strain (yjbI^-^/yjbI^WT^). The results suggest that antioxidant capacity may be restored by complemented yjbI(WT) to the extent that the biofilm in Figure 1b is maintained. Although the complemented strain was not completely rescued in HClO sensitivity as much as obtained for the parent strain, most probably due to differences in the timing and amount of yjbI expression and/or a non-specific effect by forced recombinant expression. The yjbI/yjbI^W89F^ strain showed sensitivity to the same extent as the yjbI^-^/yjbI^WT^ strain. In contrast, both yjbI^-^/yjbI^Y25F^ and yjbI^-^/yjb^Y63F^ strains did not recover HClO resistance, which was identical to that of the yjbI-deficient mutant. The apparent discrepancy between the biofilm water repellence and the HClO sensitivity of yjbI^-^/yjbI^Y25F^ strain may be due to differences in the experimental conditions, such as the levels of oxidative stress and the type of growth (biofilm or planktonic). To explain these observations, we have added the following text to the revised manuscript (lines 230–240):

“Furthermore, HClO resistance was partially restored by the introduction of a plasmid-encoded wildtype yjbI gene (yjbI^WT^) into the yjbI-deficient strain (Table 2). Among the mutant strains complemented with the yjbI derivatives, yjbI^-^/yjbI^W89F^ strain recovered its HClO resistance to the same extent as the yjbI^-^/yjbI^WT^, whereas both yjbI^-^/yjbI^Y25F^ and yjbI^-^/yjbI^Y63F^ mutants showed high sensitivity to HClO, similar to the yjbI-deficient mutant. The apparent discrepancy between the biofilm water repellence and the HClO sensitivity of yjbI^-^/yjbI^Y25F^ may be due to differences in experimental conditions, such as the levels of oxidative stress and the type of growth (biofilms or planktonic). To determine whether the significant phenotypic changes observed for yjbI^-^/yjbI^Y63F^ relate to the protein hydroperoxide peroxidase-like activity of YjbI, we purified the Y63F derivative (Figure 4—figure supplement 1) and evaluated its ability to remove the hydroperoxide groups in BSA-OOH.”

In this connection, the following texts have been added to the Methods section of the revised manuscript in lines 148-149 and 515-517:

“and the planktonically grown culture supernatants”.

“Tetracycline was not added to the pre-culture and evaluation solid media in the yjbI^-^/yjbI^WT^ and pointmutated strains to ensure that tetracycline did not cause oxidative damage synergistically with HClO.”

(2) Is there a reason to think that YjbI is somehow tethered to the cell surface rather than simply secreted? The manuscript text seems to suggest that it is surface localized, but none of the experiments formally show the localization. Either the wording should be modified, or a test for localization should be performed. One relatively straightforward test would be immunofluorescence microscopy of whole cells, grown planktonically or in a pellicle, using their anti-YjbI Ab as the primary and then a fluorescent secondary Ab. Is the fluorescence localized to the cell surface or to the biofilm matrix (e.g., to TasA)?

We have followed your advice and modified the text in lines 25-26, 87, 144, and 161–162:

“or the biofilm matrix”.

“or the biofilm matrix”.

“YjbI localises to the cell surface or the biofilm matrix of *B. subtilis*”.

“Taken together, these results suggest that YjbI localises to the cell surface or the biofilm matrix of *B. subtilis*.”

(3) The significance of the paper would be considerably broadened with at least one other example of YjbI importance or function in another species or even in another strain. The authors state in their rebuttal letter that such studies are underway and will be reported separately. I am sensitive to that argument, especially as some of the other teHbO-encoding species might be difficult to manipulate genetically. But what about moving the yjbI::erm mutation into strain 3610 via phage transduction, which would be trivial? Does the mutation have an even more severe effect in a robust biofilm-former, or does the relatively low biofilm-forming ability of 168 sensitize it to loss of YjbI activity?

We agree that the suggested experiments will broaden the significance of this study. Furthermore, we have already confirmed that purified *M. tuberculosis* trHbO prevents oxidative damage to proteins; therefore, we believe that trHbO might have a similar physiological function across species. However, because further experiments will require more time and effort, we respectfully report these studies separately. Although this study was based on *B. subtilis* strain 168, which is one of the most widespread bacterial strains, we believe that results from other *B. subtilis* strains should be considered in the future, together with results from different bacterial species, to develop this study. To include this aspect, we have added the following text to the revised manuscript (lines 311–314):

“Whether the YjbI orthologues found in other bacteria, including other *B. subtilis* strains, function in the suppression of oxidative protein aggregation, biofilm formation, or oxidant resistance remains an open question for future research.”